# The Neuroprotective Activities of the Novel Multi-Target Iron-Chelators in Models of Alzheimer’s Disease, Amyotrophic Lateral Sclerosis and Aging

**DOI:** 10.3390/cells12050763

**Published:** 2023-02-27

**Authors:** Lana Kupershmidt, Moussa B. H. Youdim

**Affiliations:** 1Clinical Research Institute at Rambam Health Care, Haifa 31096, Israel; 2Technion-Rappaport Family Faculty of Medicine, Haifa 31096, Israel

**Keywords:** Alzheimer’s disease, amyotrophic lateral sclerosis, oxidative stress, monoamine oxidase, iron, iron chelator, neuroprotection, erythropoietin, amyloid precursor protein, tau protein

## Abstract

The concept of chelation therapy as a valuable therapeutic approach in neurological disorders led us to develop multi-target, non-toxic, lipophilic, brain-permeable compounds with iron chelation and anti-apoptotic properties for neurodegenerative diseases, such as Parkinson’s disease (PD), Alzheimer’s disease (AD), age-related dementia and amyotrophic lateral sclerosis (ALS). Herein, we reviewed our two most effective such compounds, M30 and HLA20, based on a multimodal drug design paradigm. The compounds have been tested for their mechanisms of action using animal and cellular models such as APP/PS1 AD transgenic (Tg) mice, G93A-SOD1 mutant ALS Tg mice, C57BL/6 mice, Neuroblastoma × Spinal Cord-34 (NSC-34) hybrid cells, a battery of behavior tests, and various immunohistochemical and biochemical techniques. These novel iron chelators exhibit neuroprotective activities by attenuating relevant neurodegenerative pathology, promoting positive behavior changes, and up-regulating neuroprotective signaling pathways. Taken together, these results suggest that our multifunctional iron-chelating compounds can upregulate several neuroprotective-adaptive mechanisms and pro-survival signaling pathways in the brain and might function as ideal drugs for neurodegenerative disorders, such as PD, AD, ALS, and aging-related cognitive decline, in which oxidative stress and iron-mediated toxicity and dysregulation of iron homeostasis have been implicated.

## 1. Introduction

### 1.1. Brain-Iron Homeostasis, Oxidative Stress and Neurodegeneration

The etiology of neurodegenerative diseases is not yet well understood, although cerebrovascular atrophy that leads to brain ischemia may be a potential pathogenic factor for age-related dementia [1]. However, accumulating evidence has shown that iron-dependent oxidative stress (OS), increased iron levels, and monoamine oxidase (MAO)-B activity, as well as reduced antioxidant levels and activities in the brain, may be major pathogenic factors in neurodegenerative diseases [2]. It is well established that iron is an essential cofactor for many key proteins involved in the normal function of neuronal tissues and is normally involved in oxygen transport, storage and activation, electron transport, and many important metabolic processes [3]. In the central nervous system (CNS), iron is essential for multiple functions, including gene expression, DNA synthesis, neurotransmission, myelination, and mitochondrial electron transport (Figure 1) [4].

Iron incorporation and transport in the brain are regulated by the interaction between the endothelial cells and astrocytes: the transferrin receptor 1 (TfR1) in the luminal membrane of endothelial cells binds Fe^3+^-loaded transferrin and internalizes this complex in endosomes, where Fe^3+^ is reduced to Fe^2+^. The latter is transported to the cytosol by the divalent metal transporter-1 (DMT1) and exported into the extracellular fluid by the iron exporter, ferroportin [5]. Alternatively, it has been proposed that the transferrin-TfR1 complex may be transported from the luminal to the abluminal surface by an iron release [6]. Ceruloplasmin, expressed in the astrocyte, oxidizes newly released Fe^2+^ to Fe^3+^, which binds to transferrin in the brain interstitial fluid [5,6,7]. Fe^2+^ can also bind to adenosine triphosphate (ATP) or citrate and be transported as non-transferrin-bound iron (NTBI), which is the source of iron for astrocytes and oligodendrocytes, which do not express TfR1 [5,6].

Yet, there is increasing evidence that iron accumulation and deposition can cause a vast range of neurodegenerative disorders of the CNS [8,9,10]. Free iron can induce OS because of its interaction with H_2_O_2_ in the Fenton reaction, thus resulting in an increased formation of hydroxyl free radicals. Free radical-related OS causes molecular damage that can then lead to a critical failure of biological functions, protein modification, misfolding and aggregation and ultimately cell death [3,11,12,13].

Indeed, in aging and various neurodegenerative diseases, such as Parkinson’s disease (PD), Alzheimer’s disease (AD), Huntington’s disease (HD), amyotrophic lateral sclerosis (ALS) and multiple sclerosis (MS), iron was shown to accumulate at the site of the lesion, thus suggesting having a role in neuronal death processes [8,10,14,15,16,17]. Iron also has been shown to accumulate in the aged brain mainly in the form of ferritin, in the microglia, astrocytes, oligodendrocytes, and the various regions, including the globus pallidus, substantia nigra, putamen, caudate nucleus, dentate nucleus and frontal cortex [4].

### 1.2. Alzheimer’s Disease

AD is the most prevalent neurodegenerative disease in the elderly population, and it has been estimated that about 5% of adults over 65 years are affected by this devastating disease [18]. Its predominant clinical manifestation is progressive memory deterioration and other changes in brain function, including disordered behavior and impairment in language, comprehension, and visual-spatial skills [19]. The neuropathology of AD is characterized by several features, including extracellular deposition of amyloid β (Aβ) peptide-containing plaques in the cerebral cortical regions, accompanied by the presence of intracellular neurofibrillary tangles (NFTs) and a progressive loss of basal forebrain cholinergic neurons leading to reductions in cholinergic markers, such as acetylcholine levels, choline acetyltransferase (ChAT) and muscarinic and nicotinic acetylcholine receptor binding [20,21]. Additionally, there is accumulating evidence demonstrating that many cytotoxic signals in the AD brain can initiate apoptotic processes, including OS, inflammation, and iron accumulation [22,23,24]. Iron is significantly concentrated around amyloid senile plaques and NFTs, leading to alterations in the pattern of the interaction between iron regulatory proteins 1 and 2 (IRP1 and IRP2) and their iron-responsive element (IRE) and disruption in the sequestration and storage of iron [25,26]. Additionally, high levels of iron have been reported in the amyloid plaques of the Tg2576 mouse model for AD, resembling those seen in the brains of AD patients [27]. In addition to iron accumulation in senile plaques, it was demonstrated that the amount of iron present in the AD neuropil is twice that found in the neuropil of non-demented brains [25]. Further studies have suggested that iron accumulation could be an important contributor to the OS damage of AD pathology, and thus, the neurons in AD brains experience high oxidative load [28,29,30,31]. Indeed, it was found that NFTs and senile plaques contain redox-active transition metals and may exert pro-oxidant/antioxidant activities, depending on the balance among neuronal antioxidants and reductants [32]. Post-mortem analyses of AD patients’ brains have revealed activation of two enzymatic indicators of cellular OS: heme oxygenase (HO-1) [33] and nicotinamide adenine dinucleotide phosphate (NADPH) oxidase [34]. In addition, HO-1 was greatly enhanced in neurons and astrocytes of the hippocampus and cerebral cortex of Alzheimer’s subjects, co-localizing to senile plaques and NFTs. It was reported that ribosomal RNA provided a binding site for redox-active iron and served as a redox center within the cytoplasm of vulnerable neurons in the AD brain prior to the appearance of morphological changes indicating neurodegeneration [29]. Many studies suggested that iron homeostasis is disrupted in AD [25,35,36]. Thus, abnormal localization of the IRP1, and IRP2, which was shown in AD, might be linked to impaired iron homeostasis in AD [37]. However, this is likely to be a secondary effect via another process, such as increased HO-1 activity in response to cellular OS [38] or a decrease in heme bioavailability resulting from Aβ binding to heme, increasing free iron levels [39]. The location of the iron-transport protein, transferrin, in senile plaques, instead of its regular location in the cytosol of oligodendrocytes, may indicate that transferring becomes trapped within plaques while transporting iron between cells [35]. Indeed, Loeffler and collaborators [40] reported on diminished transferrin/iron ratios in various brain regions of AD patients compared with normal elderly controls, indicating dysregulation of iron homeostasis. In addition, higher transferrin C2 allele occurrence has been described in AD compared with normal controls [41]. Previous studies assessing the effect of certain genes coding for proteins involved in iron metabolism, such as hemochromatosis (HFE) [42] and transferring C2 gene variants, showed a high fold risk of developing AD in aged bicarriers [43]. In addition, the mediator of iron uptake by cells, melanotransferrin and the iron-storage protein ferritin, are altered in AD and are expressed within reactive microglial cells that are present both in and around senile plaques [42]. At the biochemical level, iron was demonstrated to facilitate the aggregation of Aβ and induce aggregation of the major constituent of neurofibrillary tangles (NFTs), hyperphosphorylated tau protein [28,44]. It was suggested that the toxicity of Aβ is mediated, at least in part, via redox-active iron; neuronal toxicity was significantly attenuated when Aβ was pretreated with the natural prototype iron chelator/radical scavenger, desferrioxamine (DFO), while conversely, toxicity was restored to original levels following incubation of Aβ with excess free iron [45]. In addition, previous in vitro studies demonstrated that Aβ has a high affinity for iron, and the iron-binding sites are in the hydrophilic N-terminal part of the peptide [46]. Furthermore, the presence of redox-available iron in association with pathological lesions and increased OS strongly support the notion that oxidative damage plays an important role in the pathogenesis of AD [31]. Iron regulates amyloid precursor protein (APP) translation via functional IRE. Another molecular link between iron metabolism and AD pathogenesis was provided by Rogers et al. [47], who described the presence of an IRE-type II in the 5′ untranslated region (5′UTR) of the APP transcript encoding the Alzheimer’s APP (51 to 94 from the 5′-cap site). The APP mRNA IRE is located immediately upstream of the interleukin-1 responsive acute box domain (101 to 146) [23]. Thus, APP 5′UTR is selectively responsive to intracellular iron levels in a pattern that reflects iron-dependent regulation of intracellular APP synthesis. This signifies the APP molecule as metalloproteinase, which resembles that of the iron-associated protein ferritin, a central iron-storage molecule, and the iron-regulated transporter protein-1 (IREG-1), which transports iron from enterocytes into the bloodstream [48,49,50]. Indeed, iron levels were shown to regulate mRNA translation of APP holo-protein in astrocytes [47,51] and neuroblastoma cells by a pathway like iron control in the translation of the ferritin-L and -H mRNAs by IREs in their 5′UTRs [50]. An additional study demonstrated that IRP1, but not IRP2, selectively bound the APP IRE in human neural cells [52]. Recently, Duce and coworkers [53] have identified APP as a functional ferroxidase like ceruloplasmin. Ferroxidases prevent OS caused by Fenton and Haber–Weiss reactions by oxidizing Fe^2+^ to Fe^3+^. Both holo-APP and soluble forms of APP (sAPP) species were found to have interactions with ferroportin to regulate iron transport from neurons [53]. The authors suggested that APP, by discharging neurons of Fe^2+^, may play an important role in preventing iron-mediated OS through two alternative domains: a heme-oxygenase-inhibitory domain that prevents the release of Fe^2+^ from heme and the ferroxidase domain, which oxidizes Fe^2+^ to Fe^3+^ [53]. In addition, iron has been shown to facilitate Aβ and hyperphosphorylated tau aggregation. At the biochemical level, Aβ is defined as a redox-active metalloprotein that is precipitated by interactions with neocortical metal ions, especially zinc, copper, and iron resulting in its auto-aggregation and oligomerization [54,55]. Three histidine and one tyrosine residues at the hydrophilic N terminus of the peptide are crucial for metal ion binding to the peptide [54,55]. Regarding the interaction with iron, it was found that Fe^3+^ influences the formation of amyloid fibrils of Aβ1–42 [56] and Aβ25–35 [57]. The interaction of redox-active Fe^3+^ with Aβ fibrils results in their chemical reduction and concomitant generation of reactive oxygen species (ROS), causing lipid membrane peroxidation, DNA breakdown, and protein oxidation [44,58,59,60]. Partial aggregated and oligomerized intracellular Aβ was documented to be cytotoxic and synaptotoxic in cell culture and in vivo [45,61,62,63,64]. In vitro, iron was demonstrated to enhance Aβ toxicity since the removal of iron from the culture medium or the inclusion of adequate iron chelators reduced or blocked Aβ-induced toxicity while overloading cells with iron-potentiated neuronal toxicity [45,65]. Thus, Fe^3+^ in culture media may bind to soluble Aβ and, given its amphiphilic nature, may subsequently attach to the cell membrane, thereby inducing oxidative injury [66,67]. Iron may enhance Aβ toxicity by preventing the formation of mature well-ordered fibrillar aggregation of Aβ [68]. Consequently, it has been suggested that abnormal iron deposition, such as in AD plaques, can interact with Aβ to mediate free radical-induced neurotoxicity and may account, in part, for the widespread oxidative damage in AD brains, accelerating the pathological process [31]. In addition, iron was found to accumulate in NFTs [31,69]. Fe^3+^ can bind hyperphosphorylated tau and induce its aggregation in vitro, leading to the formation of NFTs in AD.

### 1.3. Iron-Chelating Therapeutic Strategy in AD

Currently, numerous clinical trials have demonstrated the safety and efficacy of acetylcholinesterase inhibitors (AChEIs) in the treatment of AD. Yet, their benefits in AD as symptomatic drugs are likely to be more complex than a simple replacement of lost acetylcholine [70,71,72,73]. As reviewed previously, there is growing preclinical evidence that AChEIs have minor therapeutic effects and may block some of the fundamental neurodegenerative processes involved in AD [74]. Thus, much of pharmacology efforts are allocated towards other therapeutic approaches for AD, such as targeting iron neurotoxicity. Previous studies have demonstrated that metal-chelating compounds confer a potential to prevent metal-induced ROS, OS, and Aβ peptide aggregation [75]. Recently, several studies have reported a survey of various metal chelators, including restoring iron homeostasis compounds, that are potentially useful for the treatment of AD [76,77,78,79,80]. The iron chelator, DFO, has been reported in a single clinical study to slow down the progression of AD dementia [81], to prevent the formation of β-pleated sheets of Aβ1–42 and dissolve preformed β-pleated sheets of plaque-like amyloid in vitro [56]. Some clinical success has also been achieved with another metal-complexing agent, clioquinol [82]. Nonetheless, the long-term treatment of clioquinol (first generation, PBT1) caused side effects ending up with subacute myelopathic neuropathy and no significant effect on the cognition of AD patients [83,84,85]. However, a Phase IIa clinical trial reported positive results with the safe use of the second generation of clioquinol, PBT2, which significantly reduced Aβ plaque deposits and improved the cognitive behavior of AD patients [86,87,88]. In preclinical experiments, oral administration of clioquinol was reported to inhibit Aβ accumulation in an AD transgenic mouse model via its actions as a metal chelator [89], and clioquinol was recently demonstrated to chelate metal ions from metal-Aβ species and cause conformational transformation of Aβ aggregates [90]. The identification of an IRE in the 5′UTR of the APP transcript led to a novel therapeutic approach aimed at reducing amyloidosis by several FDA-pre-approved drugs targeted to the IRE in the APP mRNA 5′UTR [91,92]. For example, DFO, tetrathiomolybdate (Cu^2+^ chelator), and dimercaptopropanol (Pb^2+^ and Hg^2+^ chelator) were found to suppress APP holoprotein expression and lower Aβ peptide secretion [91,92]. In addition, the bi-functional molecule XH-1, which contains both amyloid-binding and metal-chelating moieties, was shown to reduce APP expression in SH-SY5Y cells and attenuate cerebral Aβ in APP and PS1 double-transgenic mice [93]. Additional drug classes were also reported to suppress the APP 5′UTR and limit APP expression, including antibiotics, selective serotonin reuptake inhibitors (SSRIs), and other selective receptor antagonists and agonists [92]. Up to date, available drug treatments are symptomatic with no disease-modifying effects on the underlying progressive processes in AD. A new field of therapeutic strategy is poly-pharmacology and multiple-target molecules, suggesting that drugs acting at a single target may be insufficient for the treatment of AD, which is characterized by the coexistence of several pathologies [94,95,96,97,98,99].

### 1.4. Aging

Aging of the brain has been demonstrated to be the main risk factor for AD. The association between brain aging and AD is continuously discussed. One observation holds that AD results when brain aging goes beyond a threshold. According to recent criteria, cerebrospinal fluid (CSF) Aβ changes are the primary link between AD and brain aging [100]. Evidence for the role of iron in aged-related pathologies has been demonstrated by studies showing that concentrations of non-heme iron increase in the putamen, motor cortex, prefrontal cortex, sensory cortex, and thalamus during the first 30–35 years of life [101]. Recent studies have also shown that levels of ferritin, the major iron storage protein, in older individuals were higher than in younger controls in the frontal cortex, caudate nucleus, putamen, substantia nigra, and globus pallidus [10,102]. A study comparing the cellular and regional distribution of ferritin and iron between young and aged rats has indicated that in the normal aging brain, there is an intracellular accumulation of iron in neurons [103]. A recent study involving human subjects demonstrated the correlation between iron content, as measured by quantitative magnetic resonance imaging (MRI), and cognitive impairments in elderly participants. Accordingly, the R2 MRI parameter affected by changes in brain iron concentration and water content was different in elderly participants with mild to severe levels of cognitive impairment compared with healthy controls [104], suggesting that iron misregulation might play a role in the decline in cognitive function observed in aged individuals. Recently, DFO and other metal-chelating agents have also been investigated as possible therapeutic agents for age-related neuropathology. For example, the effect of DFO was evaluated on age-related recognition memory deficits in aged Wistar rats. DFO-treated rats showed normal recognition memory in a novel object recognition task, while the saline group showed long-term recognition memory deficits. The results showed that DFO was able to reverse age-induced recognition memory deficits and reduced oxidative damage to proteins in the cortex and hippocampus, indicating that iron chelators might prevent age-related memory dysfunction.

### 1.5. Amyotrophic Lateral Sclerosis (ALS)

ALS, commonly referred to as Lou Gehrig’s disease, is a relentlessly progressive neurologic disorder with an estimated prevalence of four to six cases per 100,000 [105]. The onset of ALS is most common in midlife (usually between ages 45 and 60), with a typical disease course of 1 to 5 years. Most ALS cases (90%) are of unknown etiology and are classified as sporadic. The remaining 10% of cases are familial, and 20% of them are attributed to mutation in the superoxide dismutase-1 (SOD1) gene [106]. Both sporadic and familial forms of ALS are clinically and pathologically similar, suggesting possible common pathogenesis and the final pathway of neurodegeneration. The molecular pathogenesis of ALS is poorly understood, contributing to the lack of effective system-based therapies to treat this disease. Investigations have implied that ALS is a multifactorial and multisystemic disease that arises through a combination of several mechanisms that act by concurring damage inside motor neurons and their neighboring non-motor cells, including protein misfolding and aggregation; genetic factors; OS damage and mitochondrial dysfunction; defective axonal transport; excitotoxicity, and neuroinflammation [107,108]. In ALS patients, an imbalance in ROS production, either caused directly by mutant SOD1 or indirectly by other mechanisms, could be responsible for an altered iron homeostasis [109]. How this is accomplished is not known. An intriguing hypothesis is based on the observation that superoxide radicals, if not detoxified appropriately, can inactivate enzymes containing Fe–S clusters by oxidizing one Fe and causing its release from the cluster [110]. The increased Fe demand of the SOD-defective cells may reflect its aim to continuously reconstitute the “missing” Fe in the Fe–S clusters. An early indication of the role of iron in the pathogenesis of ALS was provided by the elevated iron levels in the CNS of both sporadic and familial forms [111,112,113]. In addition, the expression of ferritin was induced at the last stages of the disease in the SOD1-G93A transgenic mouse (which develops symptoms and pathology like those of ALS patients), indicating high iron concentrations [107]. In line with this, it was reported that transferrin is localized in Bunina bodies of spinal cord neurons from ALS patients [114,115], suggesting the involvement of transferrin in the formation of these inclusions. Interestingly, in transgenic mice expressing the wild-type SOD1 or SOD1-active mutant enzyme, G93A-SOD1, the expression of TfR and IRP1, a positive transcriptional regulator of TfR, were positively modulated in response to increased SOD1 mutation [114]. Jeong and collaborators [116] recently described the dysregulation of the iron homeostasis mechanism in the CNS in the G37R-SOD1 transgenic mice model of ALS, suggesting that iron chelation therapy might be useful for the treatment of ALS. A defect in the HFE gene, which was previously associated with iron-overload diseases, hemochromatosis, and AD, is currently associated with ALS [117]. The protein normally made by the HFE gene is thought to limit the uptake of iron by cells, protect against OS, and possibly dampen inflammatory reactions. An increased incidence of HFE mutation was reported in ALS patients [118]. The presence of this mutation was shown to disrupt the expression of tubulin and actin at the protein levels, potentially consistent with the disruption of axonal transport seen in ALS and associated with a decrease in SOD1 expression [118].

### 1.6. Therapeutic Strategies in ALS

Treatment of ALS has been fueled in part by frustration over the shortcomings of the symptomatic drugs available because these are incapable of slowing the progression of the disease and neuronal degeneration. Regrettably, the single drug approved for use in ALS, riluzole, a membrane-stabilizing drug, only slightly prolongs survival [119]. Currently, >150 different potential therapeutic agents or strategies have been tested in transgenic ALS mice, according to published trials [108]. This list involves 108 pharmacotherapies, 14 gene or antisense therapies, nine cell transplantations, three immunizations, and seven dietary or lifestyle regimens. The pharmacotherapy spectrum encompasses antioxidants, anti-excitotoxins, anti-aggregation compounds, antiapoptotic, anti-inflammatories, and neurotrophic agents. Unfortunately, therapeutic modifiers of murine ALS have failed to translate in patients successfully, probably because most of these trials tested single agents that affect only one mechanism or because of delivery limitations. Given the multiplicity of pathologic mechanisms implicated in ALS, new ALS therapies may consider a simultaneous manipulation of multiple targets. Combination treatments or polypharmacy targeting different disease mechanisms have consistently shown superior efficacy in transgenic ALS mice [120,121,122].

### 1.7. The Multifunctional Iron-Chelating Compounds

Several designed synthetic and natural multipotent compounds were investigated and described by Youdim and collaborators [17,98,123,124] to hit two or more targets implicated in AD. In a series of novel multifunctional iron chelators, the compound M30 (5-[N-methyl-N-propargylaminomethyl]-8-hydroxyquinoline) was found to be the most potent, nontoxic, lipophilic, and brain-permeable selective iron chelator (compared with zinc and copper) [123,124]. M30 and another multimodal iron-chelating compound, HLA20 (5-[4-propargylpiperazin-1-ylmethyl]-8-hydroxyquinoline) (Figure 2), were designed from the prototype brain-permeable iron chelator, VK28 (Figure 2) (Varinel Inc., West Chester, PA, USA) (5-[4-(2-hydroxyethyl) piperazine-1-ylmethyl]-quinoline-8-ol) and chemically attached to the propargyl moiety of the anti-Parkinsonian MAO-B inhibitor, rasagiline (Azilect^®^) [125] (Figure 2) thus inheriting some of their neuroprotective/neurorestorative properties [123,124,126,127,128,129,130].

In the series of multifunctional iron chelators, the compound M30 [5-(N-methyl-N-propargylaminomethyl)-8-hydroxyquinoline] (Figure 2) was found to be a most potent iron chelator, displaying highly effective inhibition of booth MAO-A and MAO-B activities, as well as iron-dependent lipid peroxidation in vitro and in vivo [123,124,128,131]. M30 possessed solubility and selective iron-chelating properties (compared with zinc and copper) [123,124] and found to be non-cytotoxic, as shown by the genotoxicity assay performed in three different cell lines, A549, SH-SY5Y, or HepG2; inhibition of cytochrome p450 isozymes and voltage-dependent potassium channel–blocking test (Varinel, Inc., West Chester, PA, USA). M30 was demonstrated to be an effective inhibitor of lipid peroxidation with higher IC_50_ value, comparable with that of the prototype iron chelator, DFO [123,124,128]. It is well established that strong iron chelators could form inert complexes with iron and interfere with the Fenton reaction, leading to a decrease in hydroxyl free radical production and thus block lipid peroxidation. M30, which has been shown to possess high iron-binding capacity [123], may also be active through this mechanism to inhibit free radical formation. In addition, M30 may act as a radical scavenger by directly blocking the formation of free radicals, as confirmed in the spin trapping of the hydroxyl radical by 5.5-dimethyl-I-pyrroline-N-oxide (DMPO), measured in the electron paramagnetic resonance (EPR) spectra (Varinel Inc., West Chester, PA, USA). It was shown that M30 could significantly reduce the DMPO-hydroxyl radical signal generated by the photolysis of H_2_O_2_ (Varinel Inc., West Chester, PA, USA).

In in vitro neuroprotective studies, M30 was found to invoke a wide range of pharmacological activities, including a neurorescue response; a protective potency against OS insults, H_2_O_2_, and SIN-1 (peroxynitrite generator, 3-morpholino sydnonimine), and a regulatory action on neuronal differentiation and neurite outgrowth in various neuronal cell lines (125,126,129) [126,127,130]. M30 was found to suppress the translation of a luciferase reporter mRNA through the APP 5′UTR sequence [127]. This effect may account, at least in part, for the observed downregulation of membrane-associated holo-APP levels in the mouse hippocampus and in SH-SY5Y neuroblastoma cells, presumably by chelating intracellular iron pools [126,127]. Furthermore, M30 markedly reduced the levels of the amyloidogenic Aβ in the medium of CHO cells, stably transfected with the APP “Swedish’’ mutation [126,127] and protected primary cultured neurons against Aβ toxicity [132]. Our recent in vitro studies in pancreatic β-cells demonstrated a decreased formation of intracellular ROS after H_2_O_2_ exposure and enhanced activity of the antioxidant detoxifying enzyme, catalase, in the protective effect of both M30 and HLA20. This cytoprotective effect was suppressed by pre-treating with a catalase inhibitor, suggesting a crucial role of catalase in the defensive action of the multifunctional iron-chelating drugs [133].

In in vivo studies, M30 was previously shown to prevent the loss of mouse tyrosine hydroxylase (TH)-positive neurons induced by post-intranigral injection of lactacystin (proteasome inhibitor); improve behavioral performances, and attenuate inhibition of ubiquitin-proteasome activity, iron increase, and microglial activation in the ipsilateral substantia nigra [134]. Moreover, we demonstrated that M30 prevented 1-methyl 4-phenyl 1,2,3,6-tetrahydropyridine (MPTP)-induced striatal dopamine depletion [128], as well as restored nigrostriatal dopaminergic neurons in the post-MPTP mouse model of PD [129,134] (Figure 3).

### 1.8. Working Hypothesis Guiding Our Research

Current therapeutic approaches suggest that drugs acting at a single target may be insufficient for the treatment of multifactorial neurodegenerative diseases such as PD, AD, and ALS, all characterized by the coexistence of multiple etiopathology (e.g., OS and ROS formation, protein misfolding, and aggregation, mitochondrial dysfunction, inflammation, metal dyshomeostasis and accumulation at the sites of neurodegeneration). Based on this reasoning, the working hypothesis of the current study was that multimodal chimeric compounds, synthesized by amalgamating the propargyl moiety of the neuroprotective/neurorestorative drug, rasagiline, into the antioxidant-chelating skeleton of an 8-hydroxyquinoline derivative of the iron chelating compound VK-28 might provide a new powerful new strategy for combating multifactorial neurodegenerative disorders.

## 2. Effects of M30 Treatment on APP/PS1 Transgenic Mouse Model of AD

### 2.1. M30 Treatment Attenuated Cognitive Deficits of APP/PS1 Transgenic Mice as Assessed by the Morris Water Maze Test

The major aim of this study was to explore whether the novel multifunctional iron-chelator, M30, exhibits beneficial effects on cognitive impairments and pathological alterations in APP/PS1 Tg mice. The effect of long-term M30 treatment (1 and 5 mg/kg for 9 months, initiated when the mice were 3 months old) on spatial learning deficits in the transgenic mice was investigated. The abilities of the mice to learn and process spatial information were evaluated by the Morris water maze test, one of the most widely accepted behavioral tests of hippocampus-dependent spatial learning and memory [135]. All mice were tested on both the visible and hidden platform versions of the Morris water maze test. Figure 4 shows the results of water maze acquisition training of all mice. The visible platform tests showed that the escape latency decreased significantly across the five days of visible platform sessions for all groups. As shown in Figure 4A, despite the significant initial spatial learning impairment exhibited by vehicle-treated APP/PS1 mice, they were able to locate the visible platform proficiently by the fifth day of the visible platform session. In the hidden platform version, vehicle-treated APP/PS1 mice showed impaired acquisition of spatial learning, compared with the non-Tg mice, as indicated by much slower improvements in the escape latency across consecutive trials. M30 treatment ameliorated the performance deficits in APP/PS1 mice during the testing period with the invisible platform, compared with the vehicle-treated group. This was followed by a probe trial performance, showing that M30 treatment not only significantly promoted the acquisition phase of place learning but also significantly improved memory retention during the probe trial (Figure 4B). Taken together, these data indicate that M30 treatment led to spatial learning-memory improvement in APP/PS1 mice.

### 2.2. Y-Maze Spontaneous Alteration

Y Maze spontaneous alteration is a particular test of memory function, navigation behaviors, and the willingness of rodents to explore new environments. Y maze has been shown to be extremely sensitive to hippocampal damage, as well as many other parts of the brain (e.g., septum, basal forebrain, and prefrontal cortex) have been shown to be involved in this task. Y-maze task revealed that vehicle-treated APP/PS1 mice were impaired in this task, compared with the non-Tg mice, while APP/PS1 mice given M30 (5 mg/kg) showed a significantly higher proportion of spontaneous alternations (Figure 5). There was no significant difference in the general activity, measured as the total number of arm entries between the groups (data not shown).

### 2.3. Hebb–Williams Mazes

In the next experiment, the effect of M30 on cognitive function was examined by the Hebb–Williams maze test. The Hebb–Williams maze is an incentive-based exteroceptive behavioral model useful for measuring the spatial working memory of rodents. Each mouse was trained for two sessions per day, three trials each, in practice maze A (Figure 6A). Each mouse was run through the practice maze until reaching the criterion of completing the 3-trial session in less than a total of 60 s for two consecutive sessions. The test phase began after all mice met the acquisition criterion. Our results revealed that, compared to non-Tg counterparts, the APP/PS1 vehicle-treated mice were impaired in the total number of errors, initial entry errors, and repeat errors in both (number 8 and 12) problems performed (Figure 6A). M30-treated mice at both concentrations showed improved performance in both problems, making significantly fewer total and repeat errors, as compared with vehicle-treated Tg controls (Figure 6B).

### 2.4. Novel Food Neophobia Test

The novel taste neophobia test that is sensitive to the amygdala and hippocampal damage was used as a measure of anxiety and memory for a novel food. Figure 7 shows that in the vehicle-treated APP/PS1 group, food intake was significantly reduced after the initial exposure compared to the non-Tg mice. M30 (1 and 5 mg/kg)-treated APP/PS1 mice consumed significantly more novel food during the second encounter compared with vehicle-treated Tg mice.

### 2.5. Nest Construction Test

The nest construction test permits an evaluation of the effect of experimental therapies on reversing apathy, as well as deficits in planning and multistep problem-solving. It is considered to reflect a type of step-by-step planning analogous to dysexecutive symptoms seen in AD [136] and, like human cases, is sensitive to lesions of the prefrontal cortex [137]. In addition, this test may reflect apathy, the most common neuropsychiatric symptom reported among individuals with AD, manifested by a lack of interest in surroundings, social withdrawal, and a loss of motivation to improve or work [138]. In the present study, the nesting behavior test, followed by the analysis of nesting scores, revealed that nesting was impaired in vehicle-treated APP/PS1 mice in comparison with the non-Tg group. A significantly improved nesting was observed in M30-treated APP/PS1 mice at both given concentrations (1 and 5 mg/kg) (Figure 8).

### 2.6. Effect of M30 Treatment on Non-Cognitive Behavior of APP/PS1 Mice as Assessed by Rotarod and Screen Tests

The effect of M30 on the non-cognitive behavior of APP/PS1 mice was analyzed by the rotarod task, routinely used to study motor coordination and balance, and screen tests, as an indicator of general muscle strength [139]. The results of both tests revealed that vehicle-treated APP/PS1 mice displayed similar performance when compared to non-Tg mice (data not shown). In addition, as mentioned above, APP/PS1 vehicle-treated mice were able to locate the visible platform by the end of the visible platform session and demonstrated similar general activity during Y-maze testing as non-transgenic mice. Therefore, the reduced spatial learning ability of vehicle-treated APP/PS1 mice was not caused by motor or sensory deficiencies. Chronic M30 treatment at both concentrations (1 and 5 mg/kg) also had no effect on both rotarod and screen tests performances of APP/PS1 mice versus vehicle-treated APP/PS1 mice (data not shown), indicating that the observed effects of M30 on cognitive-based tasks presumably cannot be due to non-cognitive effects of M30 on sensorimotor function.

### 2.7. Body and Brain Weights

Weekly body weight monitoring showed that vehicle-treated APP/PS1 mice were consistently and significantly lower in body weight gain than the non-Tg mice. Compared to vehicle-treated APP/PS1 mice, M30 (1 and 5 mg/kg)-treated APP/PS1 mice started to gain more body weight at 6 months of age, and this trend was observed for the rest of the experiment (Figure 9). At the end of the experiment, body weights of M30 (1 and 5 mg/kg)-treated APP/PS1 mice were slightly higher than those of vehicle-treated APP/PS1 mice (33.2 ± 1.6 g and 31.8 ± 1.7 g, respectively, vs. 30.6 ± 0.8 g), like the non-Tg mice (32.8 ± 1.4 g). Relative brain weights did not significantly differ between the non-Tg and APP/PS1 vehicle-treated mice (1.9 ± 0.11% vs. 1.8 ± 0.09%) and were not significantly affected by M30 treatment (1.8 ± 0.1% and 1.9 ± 0.12% for 1 and 5 mg/kg-treated groups, respectively).

### 2.8. Effect of M30 on Cerebral Iron, Plaque Deposition, and Aβ Levels in APP/PS1 Mice

After the behavioral assessment, we studied the effect of M30 on various pathological features of AD, including cerebral iron levels, changes in fibrillar amyloid deposition, and Aβ levels in the brain. A qualitative examination of Perl’s-DAB-stained brain sections revealed an increase in iron concentration in vehicle-treated APP/PS1 mice (n = 4) as compared with virtually undetectable levels in vehicle-treated non-Tg mice (n = 3) (Figure 10). The iron levels were more pronounced in the striatum (1.9 ± 0.21) than in the cortical and hippocampal areas (0.52 ± 0.21 and 0.29 ± 0.12, respectively), as determined by OD analysis. M30 (1 and 5 mg/kg)-treated APP/PS1 mice showed notably reduced levels of iron staining in all brain regions studied, compared with the vehicle-treated group (Figure 10). The cortical and hippocampal iron levels in M30 (1 mg/kg)-treated APP/PS1 mice (n = 3) were 0.22 ± 0.06 and 0.16 ± 0.09, respectively, representing a 57% decrease in cortical (*p* < 0.05) and a 44% decrease in hippocampal (*p* < 0.05) iron levels. There was a 24% decrease in striatal iron level, but this was not found to be statistically significant (0.22 ± 0.14 vs. 0.29 ± 0.1; *p* = 0.1). The cortical, hippocampal and striatal iron levels in M30 (5 mg/kg)-treated APP/PS1 mice (n = 3) were 0.18 ± 0.1, 0.14 ± 0.02, and 1.12 ± 0.16, respectively. This represents a 65% decrease in cortical (*p* < 0.05), a 51% decrease in hippocampal (*p* < 0.05), and a 41% decrease in striatal iron levels (*p* < 0.05).

Further biochemical and immunohistochemical studies detected fibrillar amyloid deposits in brain slices by Thioflavin S staining (Figure 11A,B) and total Aβ plaque load, including diffuse and compacted fibrillar plaques, by a specific anti-Aβ-amyloid antibody (6E10, corresponding to amino acids 1-17 of Aβ peptide) (Figure 11C,D). We found that vehicle-treated APP/PS1 mice exhibited high levels of Aβ and fibrillar load, consistent with previous observations (Figure 11) [140]. In contrast, long-term oral administration of M30 resulted in a significant decrease of Thioflavin S-positive plaque deposition (Figure 11A,B) and total Aβ plaque burden (Figure 11C,D) in the frontal cortex, hippocampus, and parietal cortex, compared with vehicle-treated APP/PS1 mice, indicating that M30 was capable to reduce both fibrillar and nonfibrillar/diffused Aβ plaques.

Consistent with these findings, high-resolution Western blot analysis showed that Aβ levels were reduced in the frontal cortex, hippocampus, and parietal cortex of the M30-treated APP/PS1 group (Figure 12).

Next, we examined the effect of M30 on the levels of cerebral Aβ in APP/PS1 mice by a sandwich ELISA. As shown in Figure 13, the M30 treatment caused a significant decrease in brain concentrations of Aβ-40 and Aβ-42 in the TBS-soluble and guanidine-soluble brain homogenates. This indicates that the reduction in Aβ levels could account for the decrease in Aβ deposition observed in M30-treated APP/PS1 mice.

### 2.9. Effect of M30 on Levels of APP and APP-C-Terminal Fragments (CTFs) in APP/PS1 Mice

We further assessed the impact of M30 on cerebral levels of the full-length APP and α- and β-CTFs of APP in APP/PS1 mice. Figure 14A shows that M30 treatment has led to a reduction in holo APP levels in all brain regions assessed (frontal cortex, hippocampus, and parietal cortex), as indicated by Western immunoblotting using the anti-APP antibody 22C11, which recognizes an epitope located between amino acids 60 and 100 in the N-terminal part of the ectodomain of APP. Similarly, immunoblot analysis using an anti-APP C-terminal (676–695) antibody showed that compared with the vehicle-treated group, M30 treatment reduced the levels of holo APP in the frontal cortex, hippocampus, and parietal cortex (Figure 14B). M30 treatment also caused a significant reduction in the CTFs of APP, produced by α- and β-secretases, C83 and C99, respectively (Figure 14B). These results complemented the decrease in Aβ levels.

### 2.10. M30 Treatment Regulated Phosphorylation of APP, Tau, CDK5, GSK-3β and AKT

We explored a possible effect of M30 treatment on phosphorylation levels of APP and tau in the brain of APP/PS1 mice, using specific antibodies against phospho-Thr-668 of APP and phospho-tau at Ser-202. As shown in Figure 15 and Figure 16, a significant reduction in phospho-APP (Thr-668) and phospho-tau (Ser-202) was observed in the frontal cortex, hippocampus, and parietal cortex of M30-treated APP/PS1 mice, compared with the vehicle-treated group. Given the importance of CDK5 and GSK-3β/AKT in the regulation of APP, as well as the phosphorylation of tau, we determined the effect of M30 treatment on the phosphorylation levels of these kinases in the brain of APP/PS1 mice. Increased phosphorylation of Ser-9 in GSK-3β reflects the decreased activity of GSK-3β, whereas phosphorylation of AKT at Ser-473 and CDK5 at Ser-159 reflects the increased activity of AKT and CDK5. Quantification of Western blots revealed that M30 (5 mg/kg) treatment significantly increased the ratio of phospho-GSK-3β (Ser-9)/GSK-3β and decreased phospho-AKT (Ser-473)/AKT and phospho-CDK5 (Ser-159)/CDK5 ratios, compared with vehicle-treated APP/PS1 mice (Figure 15 and Figure 16). The levels of total GSK-3β, AKT, and CDK-5 were unchanged by M30 treatment (Figure 15).

### 2.11. Effect of M30 Treatment on MAP2 Immunoreactivity

The effect of M30 treatment on brain levels of microtubule-associated protein 2 (MAP2, a marker for neuronal cell bodies and dendrites) [140] was examined in the brains of APP/PS1 mice. Since global neocortical neuronal loss is not apparent in this mouse model at this age, and only local neuronal loss in the hippocampal region has been observed [140], we performed the immunohistochemical analysis in the dentate gyrus, CA1, and CA3 hippocampal areas. Consistent with previous reports on the APP/PS1 mouse model of AD [140,141], we found a marked reduction in the MAP2 immunoreactivity in the CA3 hippocampal region (Figure 17) but only an insignificant decrease in dental gyrus and CA1 (data not shown). However, in M30-treated APP/PS1 mice, significant preservation of MAP2 expression was observed in CA3 regions, accompanied by a significant improvement of the integrity of the neuronal fibers and increased neuronal body volume as compared with the vehicle-treated APP/PS1 mice. These data indicate that M30 treatment might decrease the rate of neuronal degeneration in the APP/PS1 mouse model.

## 3. Effects of M30 Treatment on Aged Mice

### 3.1. M30 Treatment Attenuated Age-Related Behavioral Deficits as Evaluated by Modified SHIRPA Analysis

To investigate the possibility that the novel multifunctional brain permeable iron-chelator, M30, could attenuate age-related cognitive deficits and β-amyloid deposition, the drug was chronically administrated by oral gavage to 15 months-old mice at concentrations of 1 and 5 mg/kg 4 times a week for 6 months. It was found that at the end of chronic M30 treatment, the average body weight was not significantly different between M30 (1 and 5 mg/kg)- and vehicle-treated mice (28.21 ± 1.25 g and 30.51 ± 2.25 g vs. 29.21 ± 2.1 g, respectively).

To analyze the effect of M30 on a broad behavioral profile, we used the modified SmithKline Beecham, Harwell, Imperial College, Royal London Hospital, Phenotype Assessment (SHIRPA) analysis, a basic semi-quantitative behavioral and functional battery that includes measures of muscle function and cerebellar, sensory, neuropsychiatric, and autonomic performances [142,143]. No significant differences were observed between M30- and vehicle-treated mice in the majority of the SHIRPA variables related to general behavior, motor control, muscle tone, reflexes, cerebellar, sensory, and autonomic functions (data not shown). However, M30 significantly reduced the levels of anxiety and aggression, as compared with vehicle-treated aged mice in a dose–response manner (Table 1). Specifically, M30 at 1 mg/kg significantly affected two anxiety-like measures, and M30 at 5 mg/kg significantly decreased five measures of neuropsychiatric functions in aged mice compared to vehicle-treated aged mice (Table 1).

Data are expressed as median followed by score range in parentheses. Results are presented only for those SHIRPA tests in which significance was noted. Shaded boxes represent statistically significant differences (*p* < 0.05) among M30 (1 or 5 mg/kg)-treated and vehicle-treated aged mice.

### 3.2. Novel Object Recognition Memory Nest Building Tasks

In the object recognition test, vehicle-treated aged mice showed significantly lower preference towards the novel object, as compared with vehicle-treated young mice in both short-term (Figure 18A) and long-term (Figure 18B) memory retention tests. However, M30 (1 and 5 mg/kg)-treated animals exhibited a significantly higher preference in exploring the novel object during the short-term (Figure 18A) and the long-term (Figure 18B) memory retention trials than vehicle-treated aged mice, as their recognition index was significantly higher than the vehicle-treated aged group.

### 3.3. Nest Building Task

Nesting behavior studies, followed by the analysis of nesting scores, revealed that nesting was impaired in vehicle-treated aged mice in comparison with vehicle-treated young mice (Figure 18). A significantly improved nest behavior has been observed in aged mice at both given concentrations of M30 (1 and 5 mg/kg) (Figure 19).

### 3.4. Open-Field Behavior Test

Results for open field exploration behavior in aged mice treated with vehicle or M30 (1 and 5 mg/kg) demonstrated that M30 at both concentrations given did not affect the number of rearings (Figure 20B), latency to start locomotion (Figure 20C), or defecation (Figure 20D), compared with vehicle-treated mice. Significant induction of locomotor activity was observed in the groups treated with M30 at both concentrations, as indicated by a higher number of crossings, compared with the vehicle-treated aged group (Figure 20A).

### 3.5. Effect of M30 on Cerebral Iron in Aged Mice

After the behavioral assessment, we studied the effect of M30 on various age-related pathological alterations, including regulation of cerebral iron levels and β-amyloid plaque deposition. A qualitative examination of Perl’s-DAB-stained brain sections revealed an increase in cortical iron concentration in vehicle-treated aged mice, as compared with virtually undetectable iron levels in vehicle-treated young mice (Figure 21). M30 (1 and 5 mg/kg)-treated aged mice showed notably reduced levels of iron staining compared with the vehicle-treated group (Figure 21).

### 3.6. Effect of M30 on Cerebral Plaque Deposition

Loid plaques by Thioflavin S staining (Figure 22) and by monoclonal anti-Aβ amyloid (17-24) antibody (4G8) immunohistochemistry (Figure 23) in brains of mice that received M30 (1 and 5 mg/kg), or vehicle. Figure 22 shows a notable induction of Thioflavin S-positive plaque deposition in cortical and hippocampal areas of vehicle-treated aged mice, as compared with young mice. M30 (1 and 5 mg/kg)-treated aged mice showed significantly reduced levels of Thioflavin S staining in both cortical and hippocampal regions vs. vehicle-treated aged mice (Figure 22). Furthermore, 4G8 antibody immunoreactive Aβ deposits were significantly reduced in the frontal cortex and hippocampus of M30-treated aged mice, as compared with the vehicle-treated (Figure 23).

### 3.7. Effect of M30 on Cerebral MAO-A and -B Activities in Aged Mice

As M30 has been previously shown to be a potent irreversible brain mitochondrial MAO-A and -B inhibitor [128], we finally examined the effect of M30 administration on MAO-A and -B activities in the cerebellum of aged mice. As shown in Table 2, M30 (5 mg/kg) caused a significant inhibition of both MAO-A and -B activities in the cerebellum of aged mice, compared to vehicle-treated aged control mice.

## 4. Effects of M30 and HLA20 Treatment on NSC-34 Cells

### 4.1. Neuroprotective Actions of M30 and HLA20 against Hydrogen Peroxide (H_2_O_2_) or Peroxynitrite Ion Generator (SIN-1)-Induced Toxicity in NSC-34 Cells

We evaluated the neuroprotective effect of the novel multimodal iron chelating drugs, M30 and HLA20, against H_2_O_2_- and SIN-1-induced neurotoxicity in NSC-34 cells, a widely used motor neuron-neuroblastoma fusion line. This cell line was chosen as it expressed many of the morphological and physiological properties of motor neurons [144]. In these experiments, M30 treatment (1–10 µM) significantly reduced cell mortality induced by H_2_O_2_ (Figure 24A) and SIN-1 (Figure 24B), analyzed by an apoptotic cell death detection ELISA, based on the use of mouse monoclonal antibodies to detect free histones and fragmented DNA. A similar neuroprotective effect was obtained with HLA20 (1–10 µM) (data not shown). In these concentrations, M30 and HLA20 alone, in the absence of the neurotoxins, had no effect on cell viability relative to control.

### 4.2. Expression of the Iron Metabolism-Related Protein, Transferrin Receptor (TfR) in Response to M30 and HLA20 in NSC-34 Cell Line

TfR is known to be induced by iron chelators through IRP-mediated mRNA stabilization [145]. Thus, if M30 and HLA20 act as iron chelators, levels of TfR should be increased in response to drug treatment. As shown in Figure 25A, at 5 and 10 µM, M30 and HLA20 induced a dose-dependent increase in TfR levels in NSC-34 cells, as indicated by Western blot analysis. Immunofluorescence staining of TfR (Figure 25B) confirmed this increase, further indicating the iron chelation effect of these drugs in NSC-34 cells.

### 4.3. Effect of M30 and HLA20 Treatment on HIF-1α Regulatory Pathways

Previous studies have proposed that the protective effects of iron chelators are not exclusively the result of suppression of the Fenton chemistry. Another potential therapeutic effect of iron chelators is based on the inhibition of the iron-dependent HIF prolyl 4-hydroxylases (PHDs) that regulate HIF stability, leading to transcriptional upregulation of a cassette of protective genes [146,147]. Thus, we next tested the effect of the novel multifunctional iron chelators on HIF-1α levels in NSC-34 cells. As shown in Figure 26, the levels of both mRNA (Figure 26A) and protein expression (Figure 26B) of HIF-1α were increased in M30- and HLA20-treated NSC-34 cells, indicating that the drugs not only stabilized HIF-1α protein, but influenced its regulation at the transcriptional level. Immunohistochemical analysis confirmed these data, and further revealed that M30 and HLA20 caused not only accumulation but also nuclear translocation of HIF-1α (Figure 26C). The expressed HIF-1α in control was almost exclusively distributed in the cytosol and rarely found in the nucleus. In contrast, NSC-34 cells treated with 10 µM M30 and HLA20 for 48 h expressed high levels of HIF-1α in both the cytosol and the nucleus. In addition, real-time RT-PCR analysis revealed that exposure of NSC-34 cells to M30 or HLA20 significantly increased mRNA levels of VEGF (Figure 27A) and enolase 1 (Figure 27B), two known HIF-1 regulated genes [146,147,148,149,150].

It has been reported that phosphatidylinositol-3-kinase (PI3K)/AKT signaling plays an important role in regulating HIF-1α expression [149,150]. Therefore, we investigated the activation of AKT by monitoring Ser-473 phosphorylation in NSC-34 lysates. Figure 28 demonstrates that both M30 (10 µM) and HLA20 (10 µM) markedly increased the amount of phospho-AKT at 30 min drug exposure. GSK-3β is a well-characterized downstream substrate of AKT [150,151]. Previous studies have demonstrated that the phosphorylation of GSK-3β at Ser-9 by AKT results in the inhibition of GSK activity [151]. In our hands, M30 and HLA20 significantly increased the phosphorylation of GSK-3β at Ser-9 in NSC-34 cells (Figure 28), suggesting that both drugs induced activation of AKT followed by phosphorylation (inactivation) of GSK-3β.

### 4.4. Effect of M30 and HLA20 Treatment on NSC-34 Cell Differentiation: Molecular Mechanisms

Next, we demonstrated that while untreated control cells appear to be characterized by short cell processes, NSC-34 cells treated with M30 (2.5–10 µM) or HLA20 (2.5–10 µM) were characterized by a higher number of cell processes, displaying a marked neuron-like phenotype (Figure 29). These neurites appear to be very well extended from the cell body, with a mean length ranging from 5–10 times the soma size. The morphological modifications were accompanied by a significant increase in the expression of the neuronal-specific axonal marker of differentiation GAP-43, as determined by immunofluorescence (Figure 29A,B), Western blot analysis (Figure 29C), and real-time PCR (Figure 30A). Moreover, treatment with M30 (10 µM) and HLA20 (10 µM) appeared to significantly increase mRNA expression of BDNF (Figure 30B), consistent with the presence of both pro- and mature forms of BDNF in NSC-34 cell lysates [152] and with a possible autocrine role of BDNF for motor neurons. These results are in line with previous studies implicating the importance of iron in DNA replication, and indeed, cellular depletion of iron has been shown to induce cell differentiation and cell-cycle regulation [153,154,155]. We further examined the effect of M30 and HLA20 on the levels of cyclin D1, which plays a critical function in G1 progression by interacting with cyclin-dependent kinases and can be regulated post-transcriptionally by iron-depletion [155]. Figure 31 shows that both M30 and HLA20 at concentrations of 5 and 10 µM, markedly reduced the levels of cyclin D1, as determined by Western immunoblotting, suggesting an inhibitory effect of these iron-chelator compounds on cell cycle reentry [127].

As many reports have suggested the involvement of extracellular signal-regulated kinases (ERKs) in the neuronal differentiation [156,157], we examined whether ERKs may play a role in M30- and HLA20-stimulated neurite outgrowth. To block the activation of ERKs, we used PD98059, a selective inhibitor of MEK. PD98059 non-competitively blocks the activation of MEK by Raf-1 without affecting other known serine/threonine and tyrosine kinases [158]. Figure 31 demonstrates that pre-treatment of NSC-34 cells with PD98059 (10 µM) almost completely abolished M30- and HLA20-induced neurogenesis. In addition, prior exposure to the PKC-specific inhibitor, GF109203X (2.5 µM), significantly attenuated neurite outgrowth induced by M30 and HLA20, indicating the involvement of both ERK and PKC activation (Figure 32). Indeed, M30 (10 µM) (Figure 33A) and HLA20 (10 µM) (Figure 33B) caused rapid phosphorylation of PKC and ERK1/2 in NSC-34 cells at 10 min, reaching a maximum of 20 min.

### 4.5. Protective Effect of M30 and HLA20 against G93A-SOD1-Induced Toxicity in NSC-34 Cells

Nonconfluent cultured G93A-SOD1 NSC-34 cells were shifted into a growth medium containing 1 mg/mL doxycycline to induce mutant G93A-SOD1 expression 48 h before administration of the drugs for a further 48 h. Figure 34 shows that cell viability was markedly reduced after induction of mutation (48 h), compared to empty vector-transfected NSC-34 cells, while M30 and HLA-20 (5 and 10 µM) significantly increased G93A-SOD1 NSC-34 cell viability, as determined by the MTT reduction analysis.

## 5. Effects of M30 Treatment on ALS Transgenic Mouse Model

### Effects of M30 Treatment in G93A-SOD1 Mutant ALS Transgenic Mice

The positive outcome of our study, conducted in motor neuron cell cultures, has encouraged us to investigate the effect of the multifunctional iron chelating compound M30 in the fast-progressing strain of G93A-SOD1 mutant ALS transgenic mice. Oral administration of M30 (1 mg/kg) four times a week, starting from the 70th day after birth and continuing until death, resulted in delaying the onset of motor dysfunction (Figure 35). The plot of cumulative probability of the symptom onset against the age of animals (Figure 35A) shows a significant shift to the right by M30 treatment, compared with the vehicle-treated group. The average age of onset was 107 ± 3 days in the control group and 112 ± 4 days in the M30-treated group (n = 14–16; *p* < 0.001; log-rank Mantel–Cox test) (Figure 35B). Next, we examined the effects of M30 treatment on mice survival and life span. Kaplan–Meyer curve illustrates an increase in survival by M30 treatment (Figure 35C) from 124 ± 6 to 134 ± 12 days (n = 14–16; *p* < 0.025; log-rank Mantel–Cox test) (Figure 35D). In addition, the effect of M30 treatment on overall deficit scores of motor dysfunction was assessed by four independent behavioral tests and plotted against the age of the ALS transgenic mice. Figure 35E shows that the curve of dysfunction scores was shifted to the right by the M30 treatment. Body weight was also evaluated in the course of disease progression. As demonstrated in Figure 35F, M30 treatment slightly attenuated the weight loss of G93A-SOD1 mice.

## 6. Effects of M30 Treatment on C57BL/6 Mice

### 6.1. Effects of M30 Treatment on Neuroprotective-Adaptive Mechanisms and Pro-Survival Signaling Pathways

Considering the observed neuroprotective ability of M30 in the transgenic mouse models of AD and ALS and age-related cognitive decline, we intended to provide further insight into the various endogenous molecular mechanisms and pro-survival signaling pathways, activated in the brain following M30 systemic administration that might mediate neuroprotection.

### 6.2. Effects of M30 Treatment on HIF-1α Levels and HIF-1-Target Gene Expression

HIF-1α, a master regulator of cellular oxygen homeostasis, is stabilized and activated by hypoxia or treatment with heavy metals such as cobalt chloride or iron chelation with DFO and deferasirox, and modulates the expression of several target genes, which could contribute to neuroprotection [125]. To investigate the systemic effects of the multifunctional iron chelator M30 on HIF-1α levels and HIF-1-dependent target genes in the brain, the drug was chronically administrated by oral gavage to adult C57BL/6 mice.

At the end of the chronic 30 d treatment with M30, the average body weight was not significantly different between M30- and vehicle-treated mice (26.8 ± 5.3 g vs. 28.1 ± 4.9 g, respectively). Serum iron levels of M30-treated (n = 12) and vehicle control mice (n = 11) were 110.5 ± 16.4 μg iron/dL and 172.4 ± 26.2 μg/dL, respectively (*p* < 0.05). Liver iron levels of M30-treated and vehicle control mice were 110.49 ± 20.6 μg iron/100 mg tissue and 145.7 ± 15.8 μg/100 mg tissue (*p* > 0.05).

Western blotting for HIF-1α and real-time RT-PCR analysis of selected HIF-1-target genes was performed in the frontal cortex, hippocampus, striatum, and spinal cord tissue samples. In agreement with previous results, HIF-1α protein levels were virtually undetectable in the vehicle controls [159]. Treatment with M30 revealed a marked induction of HIF-1α protein levels in all brain regions assayed (cortex, hippocampus, and striatum) and spinal cord (Figure 36B) with no effect on its mRNA levels (Figure 36A), indicating that the drug-mediated HIF-1α induction is primarily at the post-translational level.

Gene expression analysis revealed that M30 differentially upregulated mRNA levels of selected downstream HIF-1-related genes (e.g., vascular endothelial growth factor (VEGF), erythropoietin (EPO), Enolase-1, transferrin receptor (TfR), heme oxygenase-1 (HO-1), inducible nitric oxide synthase (iNOS), and glucose transporter (GLUT)-1)) in brain regions (frontal cortex, hippocampus, striatum) and spinal cord (Figure 36). For example, VEGF and iNOS were significantly induced in the cortex, striatum, and spinal cord; GLUT-1 in the cortex, striatum, and hippocampus; Enolase-1 only in the cortex; and HO-1 in all brain regions and spinal cord (Figure 37). These observations indicate that M30 possesses the ability to activate the HIF-1 pathway in the central nervous system (CNS) in vivo. Those differences, demonstrated in the regulatory effect M30 on HIF-1α target genes between various CNS regions, might reflect a differential pattern of PHDs tissue distribution, enzyme activity, or regulation [130,160].

### 6.3. M30 Induces Expression Levels of Neurotrophic Factors and Antioxidant Enzymes

Previous studies have implicated that among other protective mechanisms (e.g., stabilization of mitochondrial membrane; induction of pro-survival Bcl-2 protein); the neuroprotective effect of propargylamine derivatives is ascribed to induction of neurotrophic factors and antioxidant enzymes [161]. Thus, we next tested the in vivo regulatory effect of M30 on mRNA expression levels of the neurotrophic factors, brain-derived neurotrophic factor (BDNF) and glial cell-derived neurotrophic factor (GDNF), and antioxidant enzymes, catalase, superoxide dismutase (SOD)-1 and glutathione peroxidase (GPx) in various brain regions and spinal cord. Real-time RT-PCR revealed that M30 increased mRNA expression levels of BDNF in the cortex and striatum and GDNF in the hippocampus and spinal cord (Figure 38). Additionally, M30 administration resulted in a significant increase in mRNA levels of catalase in all brain regions and spinal cord, SOD-1 in the cortex and spinal cord, and GPx in the cortex and striatum (Figure 39).

### 6.4. Effects of M30 Treatment on the Brain and Spinal Cord Signal Transduction

To evaluate the regulatory effect of M30 on CNS signaling cascades implicated in neuronal survival molecular processes, Western blotting analysis was performed using a panel of specific relevant anti-phosphorylated PKC, ERK1/2, AKT, and GSK-3β antibodies. Figure 40 demonstrates that M30 treatment significantly increased the levels of phospho-PKC, -ERK1/2, -AKT (Ser-437), and -GSK-3β (Ser-9) in the cortex and striatum of mice. In addition, pAKT (Ser-437) was elevated following M30 treatment in mice hippocampus and spinal cord (Figure 40). Taken together, these results show that M30 regulates the phosphorylation status of pro-survival signaling pathways in the CNS.

### 6.5. M30 Regulates HIF-1α and HIF-1-Related Genes, Antioxidant Enzymes and Signaling Pathways in the Heart and Liver

To determine whether the regulatory effects observed following M30 treatment in adult mice CNS occur in periphery organs, we studied the effect of the drug on the expression of HIF-1α and the same HIF-1-target genes in the heart and liver. In addition, the regulation of antioxidant enzymes and signaling pathways in these periphery organs, following treatment with M30, was examined. As shown in Figure 41A, M30 administration significantly induced HIF-1α protein levels in both the heart and liver. Moreover, in response to M30 treatment, organ-specific regulation of several HIF-1-related genes and antioxidant enzymes was evident (Figure 41B,C), pointing to a common regulatory mechanism for CNS and periphery organs. Gene expression analysis in liver samples showed upregulation of VEGF, Enolase-1, TfR, iNOS, and GLUT-1 mRNA levels, while heart samples showed significant upregulation of EPO and Enolase-1. This may represent a common mechanism of drug-induced HIF-1 signaling for the brain and other organs. mRNA expression levels of SOD-1, as well as GPx, were significantly higher in the liver of M30-treated mice, and no significant effect was observed in the heart (Figure 41C). Analysis of signaling pathways revealed that in the liver of M30-treated mice, the levels of phospho-PKC, pAKT (Ser-437), and pGSK-3β (Ser-9) were significantly higher, and in the heart than those of pAKT (Ser-437) and pGSK-3β (Ser-9) (Figure 41D).

## 7. Discussion

In the present study, we have examined the potential therapeutic utility of our novel multi-target iron-chelating drug, M30, in the treatment of AD, age-related cognitive decline, and ALS. Our results suggest that the multifunctional iron chelator compounds can upregulate several neuroprotective-adaptive mechanisms and pro-survival signaling pathways in the brain and might function as ideal drugs for neurodegenerative disorders, in which oxidative stress and iron-mediated toxicity and dysregulation have been implicated.

### 7.1. Effects of M30 Treatment on Transgenic APP/PS1 AD Mouse Model

The potential therapeutic effect of M30 on AD-related neuropathology and cognitive deficits was investigated in APP/PS1 double Tg mice, a well-established AD mouse model [97,140,162,163]. The APP/PS1 mouse co-expresses a chimeric mouse/human APP containing the K595N/M596L Swedish mutations and a mutant human PS1 gene with the exon 9 deletion under the control of mouse prion promoter elements [164,165]. These transgenes co-segregate in the APP/PS1 mice, which develop AD-like cognitive deficits, as well as brain amyloid plaques, in a similar pattern to human AD, by around 6 months of age [164].

The current study demonstrates that chronic M30 treatment improved cognitive impairment and attenuated Aβ accumulation and tau phosphorylation in various brain regions of APP/PS1 Tg mice, compared with vehicle treated Tg mice. The observed beneficial response of M30 on cognitive functions may be associated with the inhibitory effect of the drug on Aβ levels and tau phosphorylation since a clear relationship has been demonstrated between Aβ accumulation and tau hyperphosphorylation and the cognitive deficits of AD Tg mouse model [166,167,168]. This inhibitory effect of M30 on Aβ levels may be attributed, at least partly, to the reduction observed in the levels of APP and APP/CTFs, which are the precursors of Aβ. Consistent with these findings, in vitro studies have previously described the regulatory effect of M30 on APP expression/processing, resulting in reduced APP expression levels and Aβ generation in SH-SY5Y neuronal cells and CHO cells, stably transfected with the “Swedish” mutation [127]. Regarding the APP processing pathway, M30 was found to activate the non-amyloidogenic pathway within the Aβ sequence, resulting in induced soluble APP, thus precluding the formation of Aβ, as also shown previously for other propargyl-containing compounds [169,170]. In addition, M30 may improve spatial memory by directly protecting the neurons from deteriorative processes. Accordingly, in vitro studies have recently demonstrated that M30 attenuated tau phosphorylation and protected cultured neurons against Aβ-induced toxicity [130]. Moreover, it has been described that M30 possesses neuroprotective/neurorescue activities, including a reduction in the pro-apoptotic proteins, Bax and Bad, and inhibition of the apoptosis-associated phosphorylated H2A.X protein and caspase-3 activation [127].

The increased level of OS in AD brain is reflected by high levels of iron, which can stimulate free radical formation (e.g., hydroxyl radicals via the Fenton reaction), enhanced lipid peroxidation, increased DNA and protein oxidation and glycation end product and decreased cytochrome C oxidase [171,172]. Thus, it is reasonable to assume that the neuroprotective action of M30 is mediated by a reduction in OS due to the iron-chelating properties of the drug. Indeed, we observed that the iron staining decreased in the cortex, striatum, and hippocampus after M30 treatment, compared with the respective brain regions in the vehicle-treated APP/PS1-treated group, indicating that the drug may prevent and/or modify the progression of neuronal degeneration by reducing excessive iron and its redox activity. It is established that iron chelators could form inert complexes with iron and interfere with the Fenton reactions, leading to a decrease in hydroxyl-free radical production, thus blocking the lipid peroxidation [10]. M30, which has been shown to possess iron-binding capacity [123], may be active through this mechanism to inhibit free radical formation. Moreover, M30 may act as a radical scavenger by directly blocking the formation of free radicals, as confirmed in the spin trapping of the hydroxyl radical by 5.5-dimethyl-I-pyrroline-N-oxide (DMPO), measured in the electron paramagnetic resonance (EPR) spectra (Varinel Inc., West Chester, PA, USA). It was shown that M30 can significantly reduce the DMPO-hydroxyl radical signal generated by the photolysis of H_2_O_2_ (Varinel Inc., West Chester, PA, USA). Studies have revealed that M30 has a lower affinity for iron than that of the prototype iron chelator, DFO, although it is highly inhibitory against iron-induced lipid peroxidation, with an IC_50_ value of 12 µM, comparable to that of DFO [123,124]. It is likely that the very high iron chelating property of DFO contributes to its cytotoxicity, which limits its application for long periods of time in pathological conditions unrelated to systemic iron overload. By contrast, a brain-permeable compound with moderate iron chelating affinity may be a more appropriate and promising agent for AD therapy in which iron is selectively accumulated in various brain regions.

Further evaluation of the effect of M30 treatment on the phosphorylation levels of APP, tau, GSK-3β/AKT, and CDK5 revealed that the drug attenuated tau phosphorylation reduced phospho-CDK5 and enhanced phospho-AKT and phospho-GSK-3β in the frontal cortex, hippocampus and parietal cortex of APP/PS1 mice. These data are consistent with our recent studies showing that M30 enhanced the AKT and GSK-3β phosphorylation pathway and attenuated tau phosphorylation in cultured cortical neurons [130]. The mechanism through which the drug modulates these kinases will require further investigation. Because increased GSK-3 activity has been previously demonstrated to be linked to spatial learning deficits in AD transgenic mice [173,174], it may be speculated that activation of the AKT/GSK-3β pathway contributes, at least partly, to the improved cognitive abilities demonstrated following the M30 treatment, in APP/PS1 mice. In addition, M30 treatment reduced the levels of phospho-APP (Thr-668), which has been shown to be upregulated in the pathological process of APP/PS1 Tg mice [140]. Previously, it has been reported that phospho-APP (Thr-668) possesses various regulatory effects on neurodegeneration and APP processing cascades, indicating that APP is mainly phosphorylated by CDK5 and GSK-3β [173,174,175,176].

Finally, we examined the effect of the drug on the expression of the neuronal marker MAP2, as previous studies have demonstrated a significant degeneration of neurons, characterized by damage/loss of neuronal fibers surrounding the plaques in APP transgenic mice [177]. Our study shows that M30 treatment attenuated the loss of the immunoreactivity of MAP2 seen in vehicle-treated APP/PS1 mice. This is concordant with previous findings that M30 has a profound impact on neuronal differentiation in SH-SY5Y and PC12 cell lines [127]. In this regard, it has been previously demonstrated that the multimodal drug, M30, and several other propargyl derivatives significantly upregulated mRNA expression of various growth factors (e.g., BDNF, nerve growth factor (NGF), and glial cell-derived neurotrophic factor (GDNF) [161,178,179,180], suggesting that the stimulation of this neuronal pathway may provide an important step in their neuroprotective activity.

### 7.2. Effects of M30 Treatment on Age-Related Neuropathology and Cognitive Decline

Since aging of the brain has been demonstrated to be the main risk factor for AD [181,182], which is one of the most prevalent neurodegenerative disorders in the elderly population, we have further examined, in the present study, the potential beneficial effects of M30 in aged mice. Our findings showed that systemic chronic treatment of aged mice with M30 had a significant positive impact on neuropsychiatry functions and cognitive age-related impairment. These beneficial responses of M30 might be attributable, at least partly, to the following mechanisms: first, given the evidence supporting a role for free radicals’ production and OS in brain dysfunction during the aging process [183,184], the neuroprotective action of M30 may be mediated by a reduction in OS, due to its iron chelating properties. Here, we showed that cerebral iron staining decreased after M30 treatment in aged mice, compared with the vehicle-treated aged group, indicating that the drug can prevent or attenuate the progression of neuronal degeneration by the reduction in excessive iron and its redox activity. Indeed, previous studies have shown that M30 is a hydroxyl radical scavenger and an effective inhibitor of lipid peroxidation with a high IC_50_ value [124]. Accordingly, a recent report has shown that the iron chelating agent, DFO, was able to reverse age-induced recognition memory deficits and reduce protein carbonylation in the cerebral cortex and hippocampus in rats, further supporting the view that age-related cognitive deficits might be associated with iron accumulation in the brain [185]. Second, M30 may beneficially influence cognitive deterioration through its effect on attenuating MAO activity. This is consistent with our previous studies showing that M30 is a potent, selective brain MAO-A and B inhibitor [128]. Like other propargylamine-containing MAO inhibitors, M30 is a potent irreversible inactivator of the enzyme and as such, it is expected to make a covalent interaction with the FAD cofactor at the active site of the enzymes [186]. Given that products of MAO-catalyzed reaction, such as aldehydes and H_2_O_2_, are major inductors of lipid peroxidation, it is assumed that activation of MAO is associated with age-related disturbances of the homeostasis and generation of free radicals in involution of the nervous tissue [187]. Indeed, it was shown that in the brain of mice, MAO-A activity remained stable between 2 and 24 months, while MAO-B activity increased significantly between 2 and 16 months [188]. Quantitative radiography studies also showed an age-related increase in MAO-B in various brain structures [189].

Previous studies have demonstrated that a series of propargylamines, including rasagiline, deprenyl, and R-2HMP, all increased superoxide dismutase (SOD) and catalase activities in several brain regions of dopaminergic nature and in systemic organs such as heart and kidney [190]. Other studies have shown that deprenyl significantly prolonged the life span of aging rats [191], although the magnitude of the effect was quite variable, depending on individual studies [192,193,194]. In the hippocampus of aged rats, rasagiline was shown to reverse several age-related mitochondrial and key regulator genes that are involved in neurodegeneration, cell survival, synaptogenesis, oxidation, and metabolism [195].

Another interesting finding of this study is the observation that M30 treatment reduced the levels of β-amyloid plaques in the cortex and hippocampus of aged mice, compared with vehicle-treated aged animals. As discussed above, this effect may be associated with the previously observed reduction in the levels of APP in CHO cells, stably transfected with the “Swedish” mutation by M30 [126,132]. It was suggested that metal chelators could reduce APP levels by modulating APP translation via an IRE in the 5′ untranslated region of the APP transcript [23,47,92,196]. Indeed, M30 was found to suppress the translation of a luciferase reporter gene fused to the APP mRNA 5′ untranslated region [127]. In addition, M30 was shown to enhance the non-amyloidogenic pathway of APP processing and increase sAPPα levels [169,170]. In aged animals, these effects may be beneficial in the face of previous findings demonstrating increased amyloid deposition early in the aging process in various species [197,198,199,200,201]. The accumulation of Aβ with aging seems to be a combination of decreased efflux transport of endogenously generated Aβ and increased influx transport from the vascular compartment, and it is likely that the proportions from each Aβ source change with time [202]. Furthermore, it has been demonstrated that the interaction of APP and beta-site amyloid cleavage enzyme 1 (BACE1) is enhanced with aging [203,204]. Increases in APP, γ-secretase, and BACE1 have also been observed in correlation with age in cells and in vivo [181,182]. Indeed, aging is viewed as the most significant risk factor for AD and is closely correlated with AD neuropathology and there is presumably a continuum in Aβ accumulation from normal aging to AD, although the mechanism underlying this transition is not yet clarified.

### 7.3. Neuroprotective Activity of Novel Multimodal Iron Chelating Drugs in Motor Neuron-like NSC-34 Cells and Transgenic G93A SOD1 Mouse Model of ALS

M30 and HLA20 were shown to possess multiple pharmacological activities in NSC-34 cells, a widely used mouse motor neuron hybrid cell line [205]. These include improvement of neuronal survival, activation of HIF-1α and induction of its expression and downstream target genes, promotion of neuronal differentiation and induction of various neuroprotective signaling pathways. We also demonstrated the ability of M30 to significantly extend the survival of G93A-SOD1 ALS transgenic mice and delay the disease onset.

Initially, we have demonstrated the protective potency of M30 and HLA20 after exposure of cultured NSC-34 cells to the OS insults, H_2_O_2_, and peroxynitrite generator, SIN-1, previously shown to be associated with motoneuron degeneration in ALS [206,207]. This observation is consistent with our previous studies showing that M30 decreased apoptosis of SH-SY5Y neuroblastoma cells when given after depriving the cells of serum support (neurorescue paradigm) via various protective mechanisms, including reduction in the pro-apoptotic proteins, Bad and Bax, and inhibition of the apoptosis-associated phosphorylated H2A.X protein (Ser-139) and caspase-3 activation [127]. Thus, considering the mechanism of action of the novel multifunctional drugs, it can be assumed that the neuroprotective effect demonstrated in NSC-34 cells may be associated with their propargyl moiety since N-propargylamine and rasagiline conferred neuroprotection/neurorescue via activation of PKC and MAPK pathways, coupled to pro-survival Bcl-2 family members and mitochondrial membrane stabilization [208,209]. On the other hand, the iron complexing moiety embedded in the drugs may favorably influence cell survival by reducing the levels of ROS and reactive nitrogen species (RNS) because of the neutralization of excessive free-reactive Fe^2+^. An alternative pathway of protection by iron chelators may include the inhibition of the iron-dependent HIF-PHD, an enzyme that regulates HIF stability. Indeed, PHDs have been suggested as an additional target for neuroprotection in various neurodegenerative diseases. Inhibition of HIF-PHDs prevents the hydroxylation and subsequent degradation of HIF-1α [146]. Stabilization and subsequent nuclear localization of HIF-1α results in heterodimerization with its partner HIF-1β, binding to the hypoxia-response element in the gene regulatory regions, and subsequent transcriptional upregulation of established protective genes, such as erythropoietin, VEGF, p21waf1/cip1 and glycolytic enzymes (e.g., aldolase and enolase 1) [146,147,148]. Siddiq et al. [146] used an in vitro model of OS to correlate the protective effects of iron chelators and small molecules and peptides that do not bind iron but do inhibit the PHDs, with their ability to activate HIF-1. This model has been further supported by the observation that DFO was neuroprotective in hippocampal neuronal culture exposed to oxygen and glucose deprivation, in addition to OS and excitotoxicity damage, while this protection was prevented by blockade of HIF-1α with antisense oligonucleotide transfection [210].

In accordance, here we showed that M30 and HLA20 induced mRNA expression of HIF-1α and enhanced the protein levels of HIF-1α and its nuclear translocation in NSC-34 cells. Previously, several in vivo studies showed regulation of HIF-1α at the transcription level, in addition to the regulation of HIF-1α by protein stabilization [147,211,212]. Further, M30 and HLA20 significantly increased the levels of the endogenous HIF-1-dependent genes, enolase 1, VEGF, and BDNF in NSC-34 cells. Although VEGF was once considered to be only a specific angiogenic factor, emerging evidence indicates that it also has direct effects on neuronal cells and protects motoneurons from cell death induced by various insults, such as oxidative stress, hypoxia/hypoglycemia, glutamate-excitotoxicity and serum deprivation [213]. Deletion of the hypoxia-responsive element in the promotor region of the VEGF gene can cause motor degeneration in mice, and low-VEGF-producing alleles of the VEGF gene are associated with motoneuron degeneration in human ALS, suggesting that VEGF is a modifier of motoneuron degeneration in human ALS [214,215,216]. Although there is evidence for and against the role of VEGF in ALS etiopathogenesis, the literature has widespread interest in developing VEGF-based therapies for motoneuron degenerative disorders, raising new hope for the treatment of ALS and other neurodegenerative diseases. Besides regulating VEGF, HIF-1 has also been shown to promote glycolytic enzyme gene expression and consequent aerobic glycolysis [217,218]. Recent studies have demonstrated that a shift in energy generation from glucose oxidation in mitochondria to aerobic glycolysis is associated with resistance to OS [219]. Thus, by inducing glycolytic enzyme expression, iron chelators may reduce the ambient free radical burden of neurons by enabling the cell to generate more energy glycolytically and minimize deleterious consequences of mitochondrial glucose oxidation. Taken together, it can be suggested that the regulation of HIF-1α expression and its related genes may constitute an additional pathway underlying the neuroprotective effect of M30 and HLA20.

Another finding of this study is the ability of M30 and HLA20 to induce differentiation of NSC-34 motoneuron cells. Both drugs were found to induce cell elongation and stimulate neurite outgrowth. These morphological modifications were accompanied by an increase in the immunoreactivity of the neuronal marker GAP-43 and a decrease in cyclin D1 expression, in accordance with results of previous studies, demonstrating that M30 induced a neuritogenic effect and triggered cell cycle arrest in G0/G1 in PC12 and SH-SY5Y cell lines [127]. Indeed, many cell cycle-regulating factors require iron for their function [155,220], suggesting that the cell cycle-blocking activity of iron chelators may trigger the process of differentiation through various iron-associated biological events [221]. In addition, the effect of the multifunctional drugs on motoneuron differentiation may be associated with their propargyl moiety since N-propargylamine and rasagiline were shown to upregulate BDNF and GDNF gene expression in PC12 cells [209,222]. Here, we show the ability of M30 and HLA20 to induce mRNA levels of BDNF, which is a well-recognized neurotrophic factor for motoneurons [223,224,225,226]. Motoneuron differentiation, induced by M30 and HLA20, was modulated by inhibitors of ERK/MAPK and PKC signaling pathways. In results complementary to inhibition studies, we found that the drugs significantly increased the immunoreactivity of phosphorylated MAPK and PKC in NSC-34 cells.

In vivo studies demonstrated that treatment with M30 provides clear benefits in G93A-SOD1-transgenic mice, significantly increasing their survival and delaying the onset of neurological dysfunction, even when the treatment was initiated at a relatively advanced stage of the disease [227]. Complementary studies conducted in the laboratory of Dr. Moussa Youdim’s collaborator, Dr. Weidong Le [228], demonstrate a significant attenuation by M30 in the elevated iron level and transferrin receptor expression, oxygen free radicals, and microglial and astrocytic activation levels in the spinal cords of the SOD1 G93A mice. These results provide further evidence that iron is involved in the pathogenesis of ALS. This may be of significant relevance for the further development of M30 for the treatment of ALS since almost all ALS patients are diagnosed after symptom onset. Considering the neuritogenic effect of M30 demonstrated in the motoneuron NSC-34 cells, it is possible that the in vivo action of M30 is mediated through the regeneration process of motor nerves, inducing neurodifferentiation and sprouting of axons, leading to the reinnervation of muscle fibers.

In conclusion, the neuroprotective/neurorescue potential of the novel rasagiline derivatives/iron-binding compounds may result from their multifunctional activities: (a) similar to rasagiline and other propargyl-containing molecules, such as ladostigil, they activate the canonical survival pathways, MEK and PKC, associated with elevation of BDNF; (b) the promotion of neurite sprouting and extension may result from activation of the above pathways, combined with the ability of iron-complexing molecules to interfere with cell cycle progression (like VK28) via deactivating cell cycle regulators, such as cyclins D1 and E, thus triggering differentiation through various iron-associated biological events. Since increased expression of the cyclin system may be involved in the mechanism of motor neuronal death at the late stage of ALS, it is reasonable to suggest that drugs directed towards cell cycle inhibition might be of value for disease treatment; (c) activation of HIF-1 and induction of its pro-survival/neuroprotective target genes (e.g., VEGF and enolase 1), an action that has been ascribed to iron chelation and inhibition of PHDs.

### 7.4. Novel Molecular Targets for M30 in Mouse Brain

Considering the observed neuroprotective ability of M30, we intended to provide further insight into the various endogenous molecular mechanisms and pro-survival signaling pathways activated in the brain following M30 systemic administration that might mediate neuroprotection.

The results in adult mice chronically treated with M30 have demonstrated that the treatment produced a significant upregulation of HIF-1α protein expression in the brain (cortex, striatum and hippocampus) and spinal cord. In addition, real-time RT-PCR revealed that M30 differentially induced the transcription of a broad range of downstream HIF-1-related protective genes within the brain, such as those involved in erythropoiesis (EPO), angiogenesis (VEGF), glycolysis (GLUT-1), and oxidative stress (HO-1), indicating a biological HIF-1 activation in the brain in response to M30 administration in vivo. This mechanism of HIF-1α upregulation is consistent with previous studies demonstrating that iron chelators may function as hypoxia mimetic regulators; stabilizing and transactivating HIF-1α, thus leading to the regulation of HIF-1-responsive genes [146,229,230,231]. This may support adaptive mechanisms, which protect the brain from a hypoxic injury through the regulation of cerebral metabolism and blood flow, promotion of angiogenesis, and induction of cytoprotection [125,160,232,233]. Previous studies have demonstrated that iron chelation by DFO enhanced HIF-1 activity and prevented neuronal death in both in vitro and in vivo models of ischemia via HIF-PHDs inhibition [132,134,229,231,234,235]. Indeed, it was found that the protective effect of DFO against neuronal death after oxygen- and glucose deprivation could be reversed by blockade of HIF-1α with an antisense oligonucleotide transfection [134]. Thus, the activation of the brain HIF-1 signal transduction pathway and consequent expression of HIF-1-target genes possessing pro-survival properties may implicate a link between M30-induced HIF-1-driven gene expression and neuroprotective capacities. In accordance, our in vitro findings demonstrated the ability of M30 to upregulate HIF-1α and several HIF-1α-target genes (e.g., enolase-1, VEGF, EPO, and p21) in cultured cortical neurons and NSC-34 cells, accompanied by protective effects against Aβ25-35- and mutant G93A-SOD1-induced toxicity, respectively [130].

Activation of the HIF-1 signaling pathway by M30 was also achieved in peripheral organs (liver and heart). For example, of the HIF-1 target genes examined in the liver, VEGF, Enolase-1, TfR, iNOS, and GLUT-1 were significantly increased. Accordingly, activation of HIF-1α was recently shown to play a role in the effect of iron depletion by DFO on glucose metabolism in hepatocytes in vitro and in vivo [236]. In HepG2 cells, DFO stabilized HIF-1α and increased the expression of GLUT1 and insulin receptors. In addition, it was shown that DFO consistently increased the phosphorylation status of AKT/PKB and its targets FoxO1 and GSK-3β, which mediate the effect of insulin on glucogenesis and glycogen synthesis and upregulated genes involved in glucose uptake and utilization. In vivo, iron depletion increased hepatic HIF-1α expression, GLUT-1 mRNA levels, and AKT/PKB activity [236]. The specific activation of HIF-1 signaling and upregulation of HIF-1-related genes in the liver may be also associated with hepatic cytoprotection, as it was shown in various models of injury that stimulation of the HIF system could protect the liver against apoptosis.

Another interesting finding is the differential upregulation of BDNF and GDNF in the CNS following M30 treatment. These data complement previous observations showing the ability of M30 and HLA20 and promote neuronal differentiation, including cell body elongation, stimulation of neurite outgrowth, and triggering cell cycle arrest in G0/G1 phase [126].

Additionally, in the current study, we showed that M30 induced mRNA expression levels of the major antioxidant defense system, comprising the detoxifying enzymes, catalase, SOD-1, and GPx, in various brain regions. The transcriptional upregulation of neuronal growth factors and antioxidant enzymes are presumably associated with the propargyl moiety embedded in the M30 molecule. Indeed, previous studies reported that several propargyl derivatives upregulated mRNA expression of BDNF and NGF and increased protein levels of BDNF [161,190,195,237], suggesting that the stimulation of these neuronal survival pathways may provide an important step in their neuroprotective activity. In line with this, it was previously shown that propargylamines possess an antioxidant action and suppress the formation of free radicals by increasing the activity of the antioxidant enzymes, SOD, and catalase in rat brain dopaminergic regions [161,190,237]. By inducing antioxidant enzymes and decreasing the formation of ROS, propargilamine-containing drugs may combat an oxidative challenge, implicated as a common causative factor in neurodegenerative diseases.

Finally, M30 treatment induced a significant increase in brain expression of phosphorylated PKC, ERK1/2, AKT, and GSK-3β. Regarding the role of these signal pathways in the regulation of neuroprotection, it has been reported in many studies that MEK/ERK and PI3K/AKT/GSK-3β pathways can promote cell survival, especially neuronal survival by both enhancing the expression of anti-apoptotic proteins and inhibiting the activity of pro-apoptotic ones [238,239,240]. In addition, these signaling pathways are well documented to play a key role in the regulation of HIF-1α [125] and, thus, might be involved in the increased expression of HIF-1α following M30 treatment. It cannot be ruled out that these signaling cascades are activated in the brain of M30-treated mice as a secondary phenomenon by a HIF-1α-dependent gene product. Thus, considering the mechanism of action of M30, it can be assumed that the neuroprotective effects of the drug may be also associated with the activation of these pro-survival signaling cascades. Indeed, N-propargylamine and rasagiline confer neuroprotection/neurorescue effect via activation of PKC and MAPK pathways, coupled with pro-survival Bcl-2 family members and mitochondrial members stabilization [125,208].

Although misregulation of the HIF pathway is only one component of a spectrum of reactions occurring in neurodegeneration, HIF-1 is a “master switch” being an important physiological response mechanism, likely resulting in several reproducible neuroprotective effects [241]. Given the wide range and diversity of cellular functions regulated by the whole spectrum of HIF-1-target genes, it is suggested that this compensatory pathway can mediate neuroprotection and is crucially involved in many physiological processes within the brain. Thus, the novel therapeutic approach of pluripotential iron-chelating compounds, such as M30, that target several pharmacological sites involved in neurodegenerative processes and activates the HIF pathway and downstream neuroprotective genes will broaden the current strategies for the treatment of neurological disorders and overall will open a new window for future development of drugs possessing a profound impact on neuron preservation.

## 8. Conclusions

Currently, simultaneous modulation of multiple targets by one multifunctional compound is the most promising approach for the multi-dysfunctional molecular conditions observed in neurodegeneration. The design of M30 and its series counterparts was to address multiple CNS etiologies in various neurodegenerative disorders, in particular AD and age-related dementia. The several targets and diverse pharmacological properties of M30 make this compound potentially valuable for a therapeutic strategy to delay neurodegeneration, as shown in the following illustration (Figure 42).

## Figures and Tables

**Figure 1 cells-12-00763-f001:**
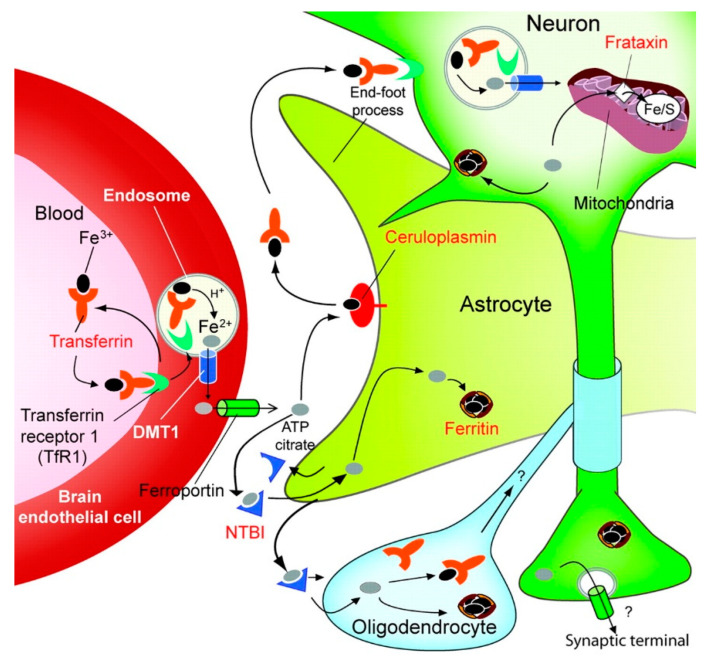
Overview of iron homeostasis in the CNS [4].

**Figure 2 cells-12-00763-f002:**
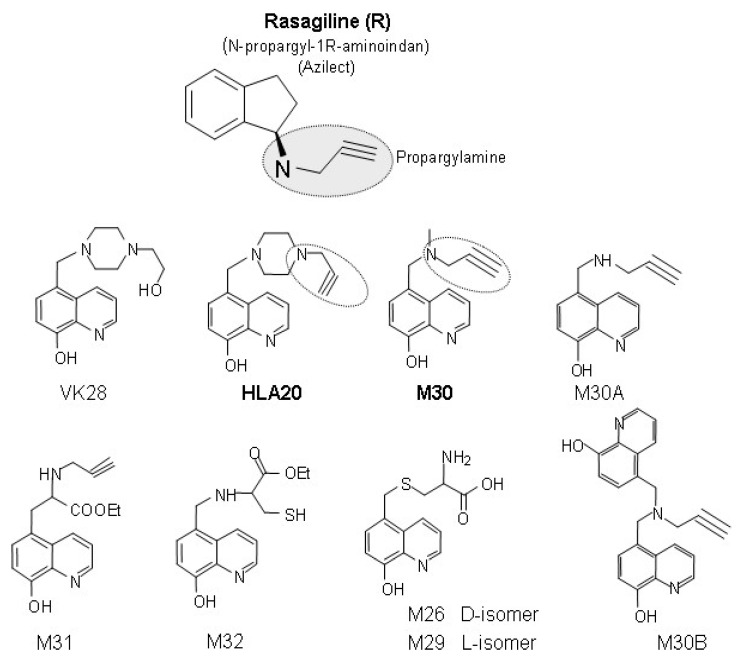
Chemical structures of novel multifunctional neuroprotective, brain permeable rasagiline-related, iron-chelating derivatives. In collaboration with Varinel Inc., which has developed one series of these compounds, exemplified by the lead iron-chelating compound called VK-28, includes new chemical entities (NCEs) that possess selective iron-chelating properties (e.g., compared with zinc and copper). Those in the second series, exemplified by multi-functional lead compounds called HLA20 and M30, possess similar iron-chelating properties, and have additional neurorestorative activity associated with the introduction of a propargylamine monoamine oxidase (MAO) inhibitor moiety of anti-Parkinson drug rasagiline (Azilect^®^). M30 is also a potent, brain-selective MAO inhibitor that inhibits both MAO-A and MAO-B in the brain while only inhibiting gastrointestinal MAOs marginally in vivo.

**Figure 3 cells-12-00763-f003:**
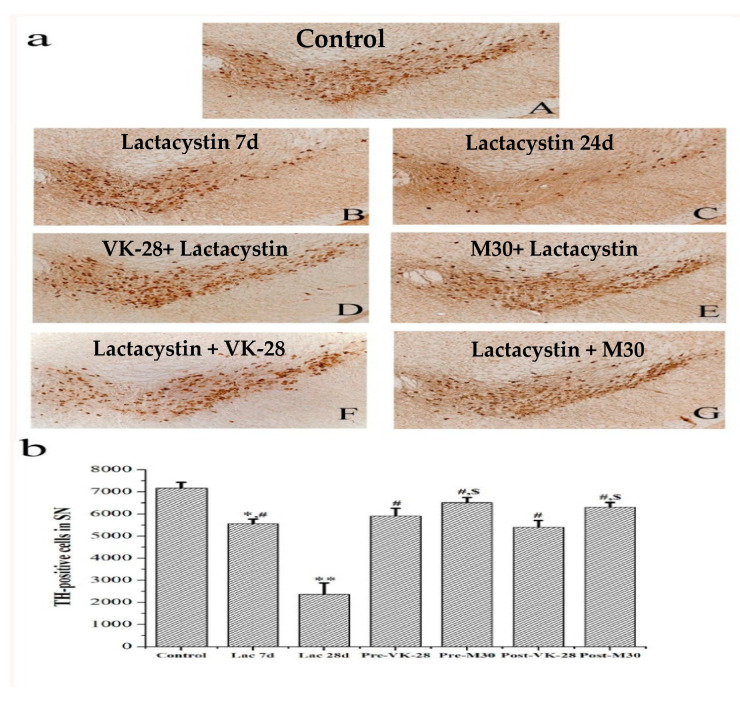
VK-28 and M30 reduced lactacystin-induced loss of DA neurons in substantia nigra (SN). Five mice injected with lactacystin (Lac) only were sacrificed 7 days after microinjection. The rest of the mice were sacrificed at the end of the study (day 28). (**a**) Representative photomicrographs of SN with TH immunohistochemistry. A–G: Control, Lac 7d, Lac 28d, Pre-VK-28, Pre-M30, Post-VK-28, and Post-M30 groups, respectively; (**b**) Quantitative analysis of TH immunopositive neurons in the SN. Each value was presented by the mean ± SEM based on the number of TH immunopositive neurons in the right hemisphericnigral slices (n = 8). * *p* < 0.01 and ** *p* < 0.001 vs. control; # *p* < 0.01 vs. Lac 28d and $ *p* < 0.05 vs. VK-28.

**Figure 4 cells-12-00763-f004:**
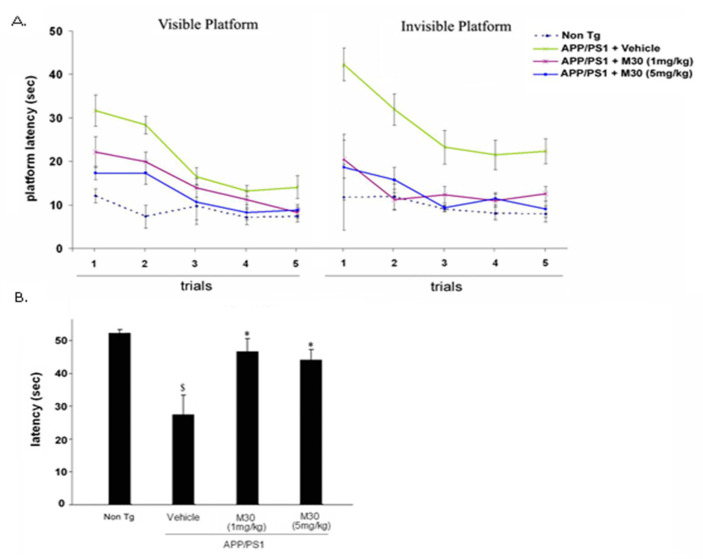
M30 treatment improved spatial learning and memory deficits in APP/PS1 mice. Morris water maze performance consisted of (**A**) escape latencies in visible-platform and invisible-platform versions; mean ± SEM, (n = 7–8); and (**B**) probe trial performance (7-day retention interval after the visible- or invisible-platform testing). Error bars represent the mean ± SEM of four trials (n = 7–8 animals in each group). $ *p* < 0.05 vs. non-Tg mice; * *p* < 0.05 vs. vehicle-treated APP/PS1 mice.

**Figure 5 cells-12-00763-f005:**
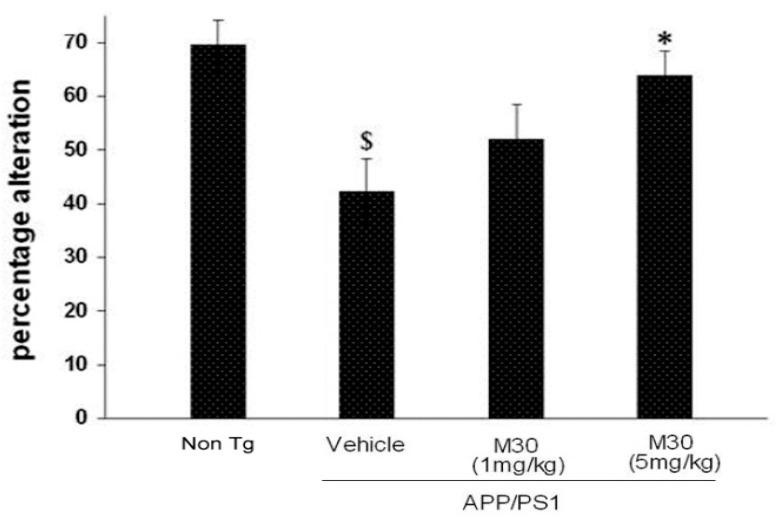
Effect of M30 (1 and 5 mg/kg) on Y-maze test performance. Alterations were counted when a mouse entered each of tree arms in succession, in any order, without re-entering one of the arms. Data are expressed as alteration percentages, calculated as the number of alterations divided by the number of total arm entries minus two during 5 min session. Each bar represents the mean ± SEM of three trials (n = 7–8 animals in each group). $ *p* < 0.05 vs. vehicle-treated non-Tg mice; * *p* < 0.05 vs. vehicle-treated APP/PS1 mice.

**Figure 6 cells-12-00763-f006:**
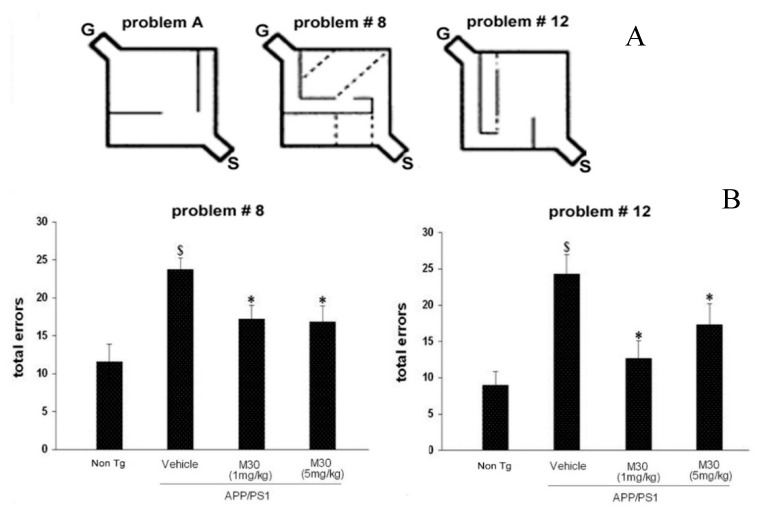
Effect of M30 (1 and 5 mg/kg) on Hebb–Williams maze test performance. (**A**). Mazes used in the study: one practice maze, problem A, and two test mazes, problems #8 and #12. For each pattern, the mouse moved from the start box, depicted in the bottom right corner, and finished when it entered the goal box located in the top left corner. Dotted lines represent the error zone; (**B**) Total errors in the test mazes #8 and #12, defined as the sum of initial (first error made in each error zone within a given trial) and repetitive (all additional errors made in the same zone) errors in 15 min session. Each bar represents the mean ± SEM of three trials (n = 7–8 animals in each group). $ *p* < 0.05 vs. vehicle-treated non-Tg mice; * *p* < 0.05 vs. vehicle-treated APP/PS1 mice.

**Figure 7 cells-12-00763-f007:**
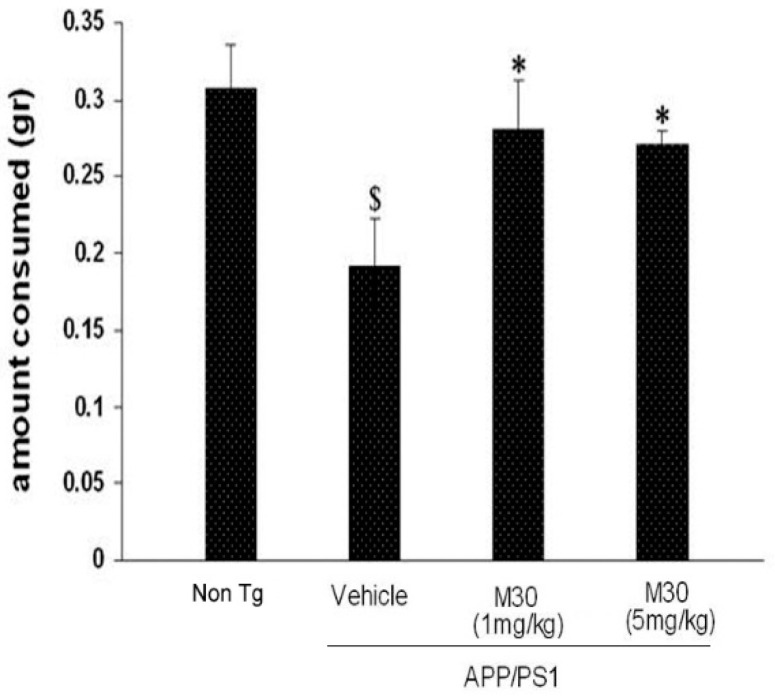
Effect of M30 (1 and 5 mg/kg) on novel taste neophobia test performance. The amount of cheese consumed by the mouse was calculated as the pre-session weight minus the adjusted for evaporation post-session weight (during 15 min session). Each bar represents the mean ± SEM of three trials (n = 7–8 animals in each group). $ *p* < 0.05 vs. vehicle-treated non-Tg mice; * *p* < 0.05 vs. vehicle-treated APP/PS1 mice.

**Figure 8 cells-12-00763-f008:**
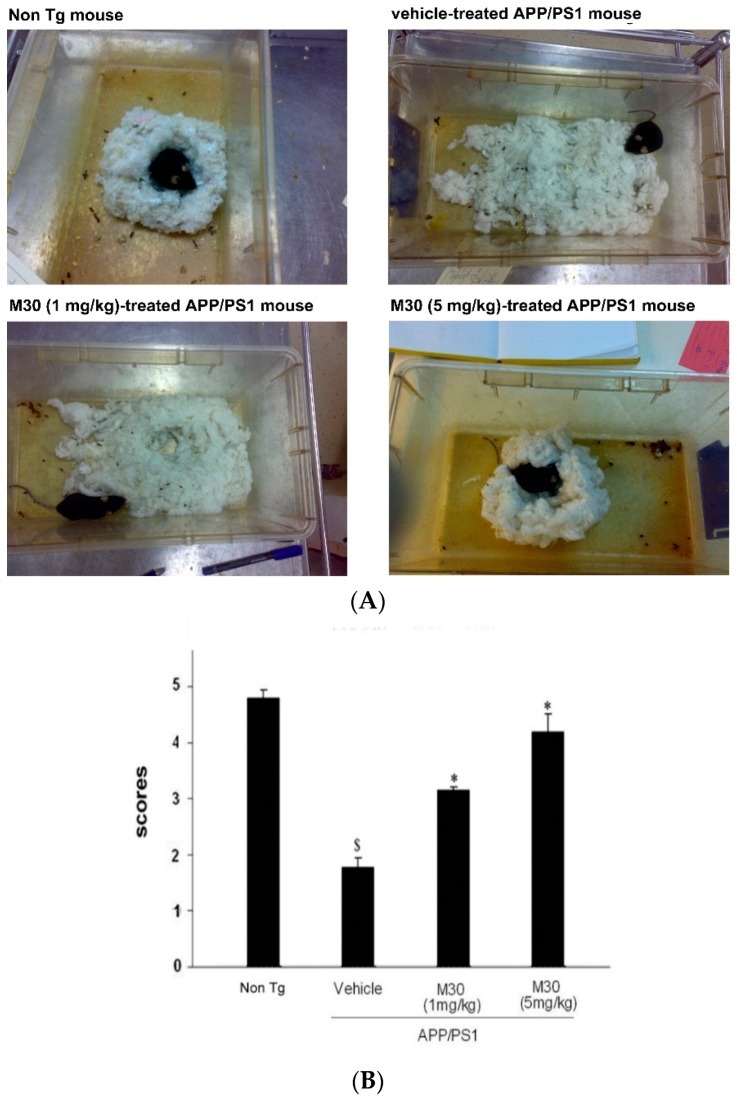
Effect of M30 (1 and 5 mg/kg) on nesting behavior. (**A**) Representative pictures of the nest constructs in four groups studied; (**B**) The presence and quality of nesting were rated on a 5-point scale as described in Materials and Methods. Each bar represents the mean ± SEM of three trials (n = 7–8 animals in each group). $ *p* < 0.05 vs. vehicle-treated non-Tg mice; * *p* < 0.05 vs. vehicle-treated APP/PS1 mice.

**Figure 9 cells-12-00763-f009:**
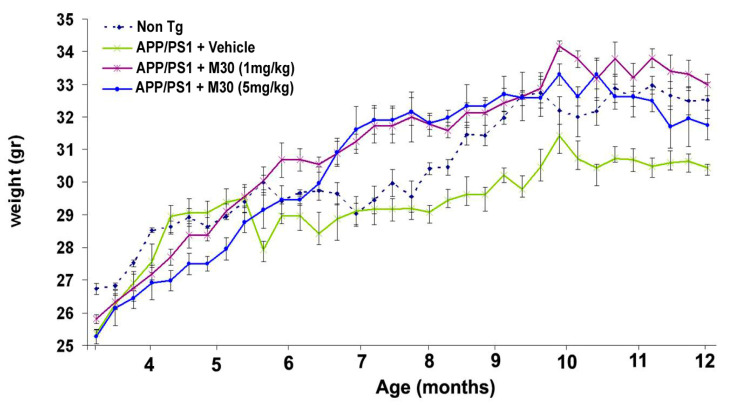
Comparison of body weight gain in non-Tg, vehicle- and M30 (1 and 5 mg/kg)-treated APP/PS1 mice. Data present weight vs. the age of animals. Values are observed by mean ± SEM (n = 7–8 animals in each group).

**Figure 10 cells-12-00763-f010:**
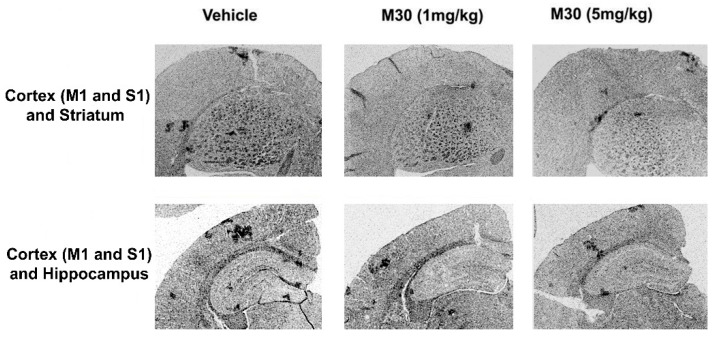
M30 treatment significantly reduced the levels of iron staining in APP/PS1 mice. Iron accumulation in various brain regions: M1 (primary motor cortex) and S1 (primary somatosensory cortex) cortex areas, hippocampus, and striatum of APP/PS1 mice treated with vehicle or M30 (1 and 5 mg/kg) were detected by enhanced Perl’s iron histochemistry.

**Figure 11 cells-12-00763-f011:**
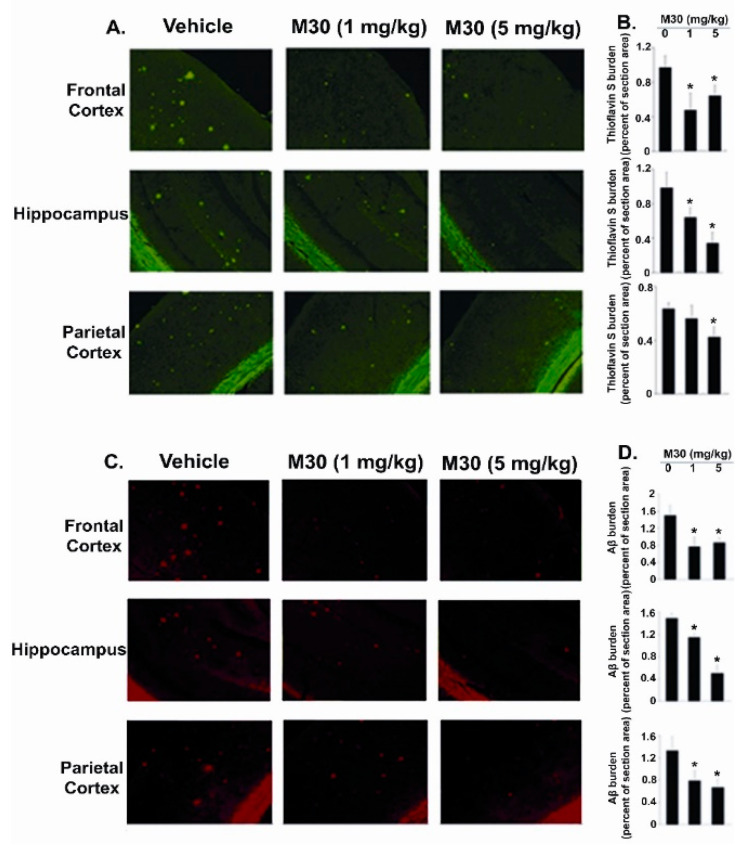
Effect of M30 treatment on cerebral amyloidosis in APP/PS1 mice. (**A**) Immunofluorescence images of mouse brain sections from the indicated regions stained with Thioflavin S; (**B**) Percentages of Thioflavin S-positive burden were calculated by quantitative image analysis; **(C**) Immunohistochemical images showing brain coronal frozen sections from the frontal cortex, hippocampus, and parietal cortex stained with β-amyloid (6E10) antibody. (**D**) Percentages of Aβ antibody-immunoreactive Aβ plaques were calculated by quantitative image analysis. Data are mean ± SEM (n = 3 animals per group; 4–8 separate fields for each animal). * *p* < 0.05 vs. vehicle-treated APP/PS1 mice.

**Figure 12 cells-12-00763-f012:**
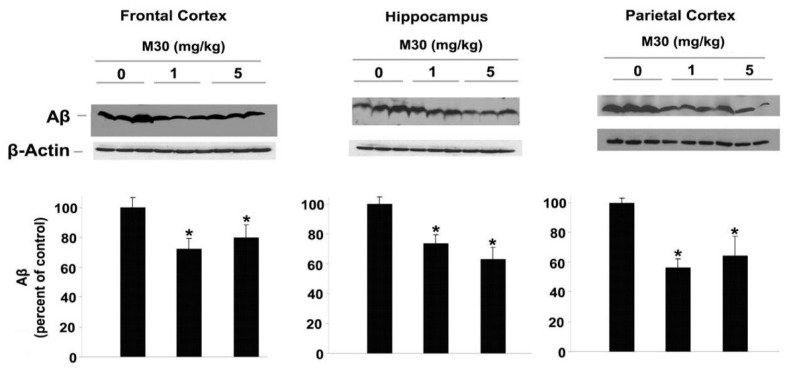
M30 treatment attenuates Aβ in APP/PS1 mice. Representative Western blots quantitative analysis of monomeric Aβ contents using β-amyloid (6E10) antibody in the frontal cortex, hippocampus, and parietal cortex lysates of APP/PS1 mice treated with vehicle or M30 (1 or 5 mg/kg). Values are normalized to levels of β-Actin and expressed as a percentage of the values from the vehicle-treated APP/PS1 mice (set to 100%) and are the mean ± SEM (n = 7–8 animals in each group). * *p* < 0.05 vs. vehicle-treated APP/PS1 mice.

**Figure 13 cells-12-00763-f013:**
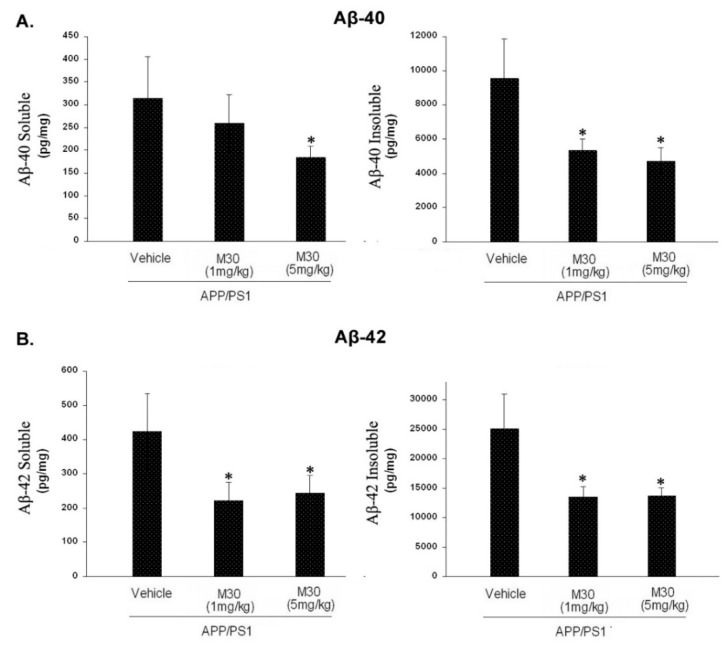
Effect of M30 treatment on cerebral Aβ-40 and Aβ-42 levels in APP/PS1 mice as analyzed by ELISA. The levels of Aβ-40 (**A**) and Aβ-42 (**B**) in the soluble and insoluble fractions were analyzed with Aβ-40 and Aβ-42 specific ELISA kits. Values are mean ± SEM (n = 7–8 animals in each group). * *p* < 0.05 vs. vehicle-treated APP/PS1 mice.

**Figure 14 cells-12-00763-f014:**
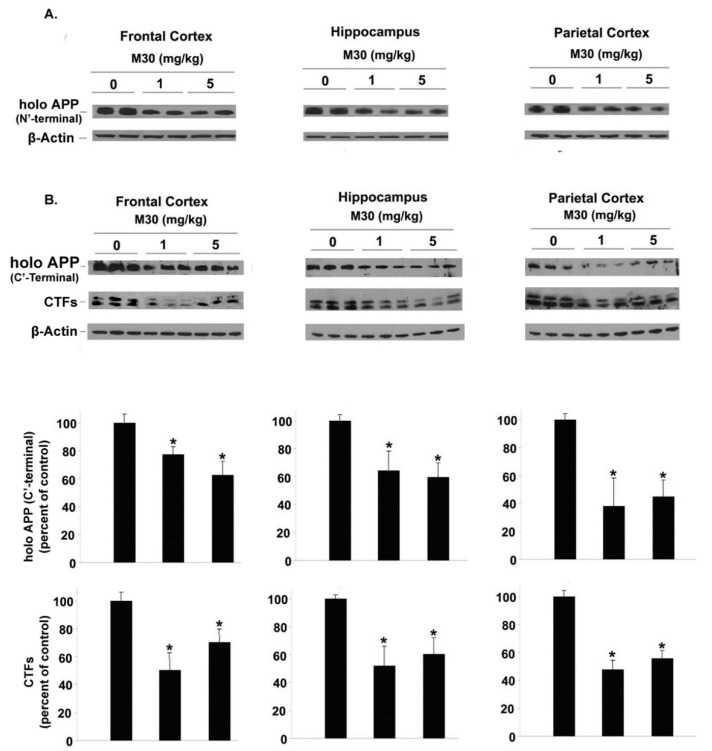
Effect of M30 on APP and APP-CTFs. (**A**) Representative Western blots and quantitative analysis of APP levels using anti-APP N-terminal specific antibody; (**B**) Representative Western blots and quantitative analysis of APP and APP-CTFs levels using anti-APP C-terminal specific antibody. Values are normalized to levels of β-Actin and expressed as a percentage of the values from vehicle-treated APP/PS1 mice and are the mean ± SEM (n = 7–8 animals in each group). * *p* < 0.05 vs. vehicle-treated APP/PS1 mice.

**Figure 15 cells-12-00763-f015:**
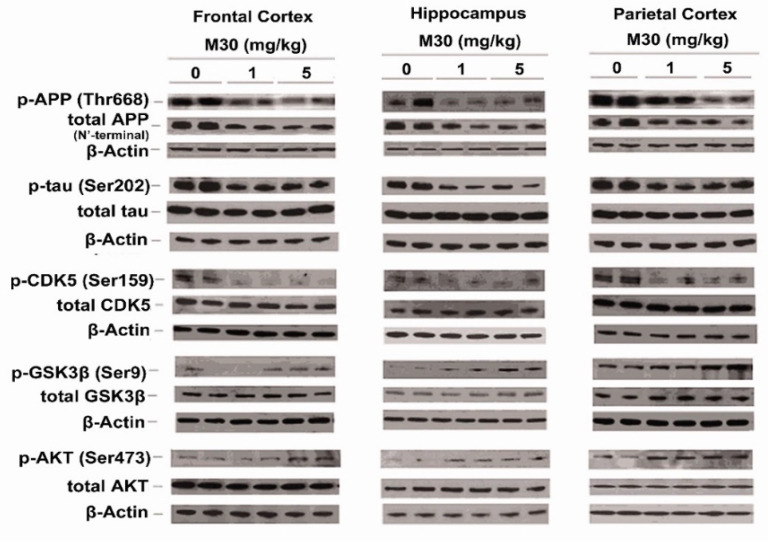
Effect of M30 treatment on phosphorylation of APP, tau, CDK5, GSK-3β, and AKT. Representative Western blots of phosphorylated APP (Thr-668), total APP, phosphorylated Tau (Ser-202), total tau, phosphorylated CDK5 (Ser-159), total CDK5, phosphorylated GSK-3β (Ser-9), total GSK-3β, phosphorylated AKT (Ser-473) and AKT in cortical (frontal and parietal) and hippocampal lysates of APP/PS1 mice treated with vehicle and M30 (1 or 5 mg/kg).

**Figure 16 cells-12-00763-f016:**
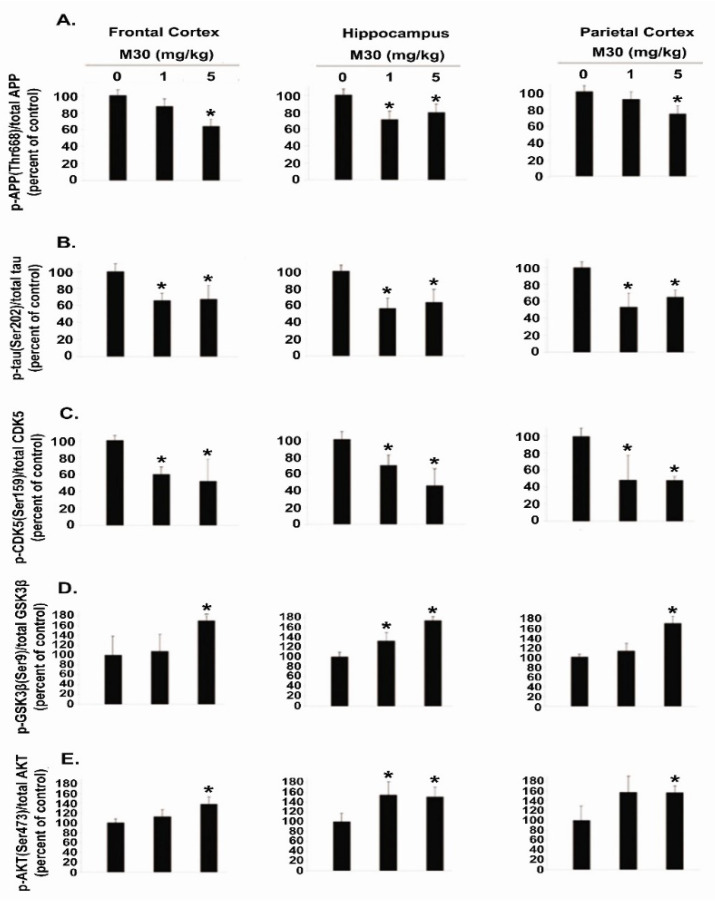
Quantitative analysis of (**A**) phosphorylated APP; (**B**) phosphorylated tau; (**C**) phosphorylated CDK5; (**D**) phosphorylated GSK-3β; and (**E**) phosphorylated AKT, normalized to levels of total tau, CDK5, GSK-3β, and AKT, respectively. In all experiments, quantified results were further normalized to β-Actin expression. Values are expressed as percentages of the values from the vehicle-treated APP/PS1 mice (set to 100%) and are the mean ± SEM (n = 7–8 animals in each group). * *p* < 0.05 vs. vehicle-treated APP/PS1 mice.

**Figure 17 cells-12-00763-f017:**
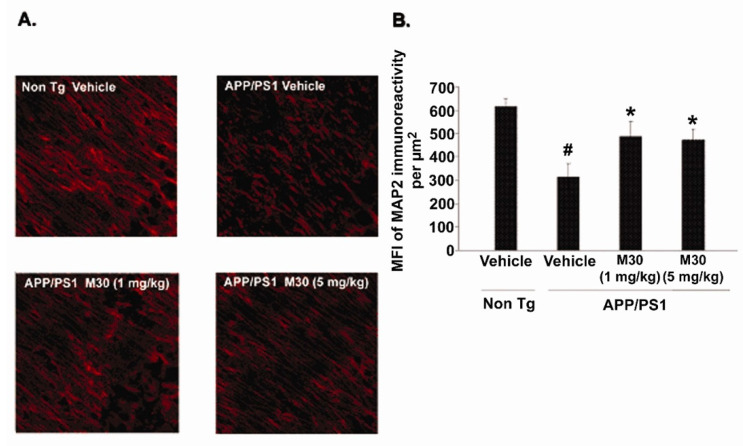
Effect of M30 treatment on MAP2 levels. (**A**) Fluorescent MAP2 protein immunostaining in the hippocampal CA3 regions of non-Tg and APP/PS1 mice treated with vehicle or M30 (1 and 5 mg/kg); (**B**) Quantification of MFI of MAP2 immunoreactivity. Values are expressed as the mean ± SEM (n = 3–4 animals per group; 4–8 separate fields for each animal). # *p* < 0.05 vs. vehicle-treated non-Tg mice; * *p* < 0.05 vs. vehicle-treated APP/PS1 mice.

**Figure 18 cells-12-00763-f018:**
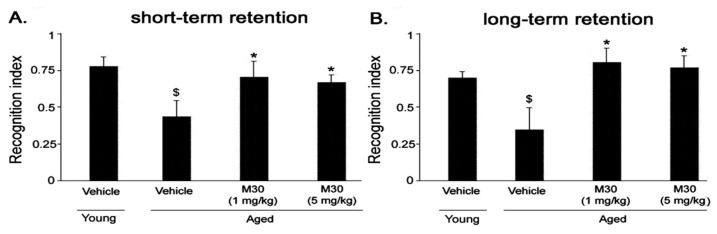
Effect of M30 on memory acquisition in aged mice. Aged mice were treated with M30 (1 and 5 mg/kg) or vehicle and subjected to the object recognition task. (**A**) Short-term (1.5 h after training); (**B**) long-term (24 h after training) retentions were assessed as described in Material and Methods. Each bar represents the mean ± SEM (n = 7 animals in each group). $ *p* < 0.05 vs. vehicle-treated young mice; * *p* < 0.05 vs. aged vehicle-treated aged mice.

**Figure 19 cells-12-00763-f019:**
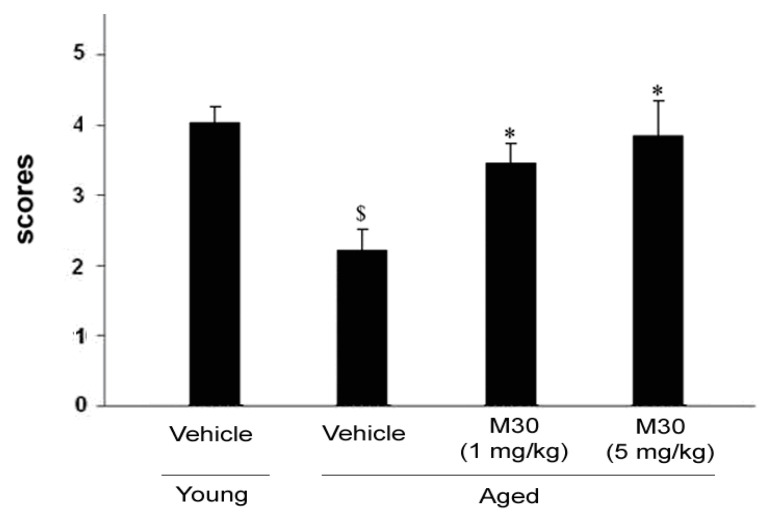
Effect of M30 on nesting behavior in aged mice. The presence and quality of nesting were rated on a 5-point scale, as described in Material and Methods. Each bar represents the mean ± SEM of three trials (n = 7 animals in each group). $ *p* < 0.05 vs. vehicle-treated young mice; * *p* < 0.05 vs. aged vehicle-treated mice.

**Figure 20 cells-12-00763-f020:**
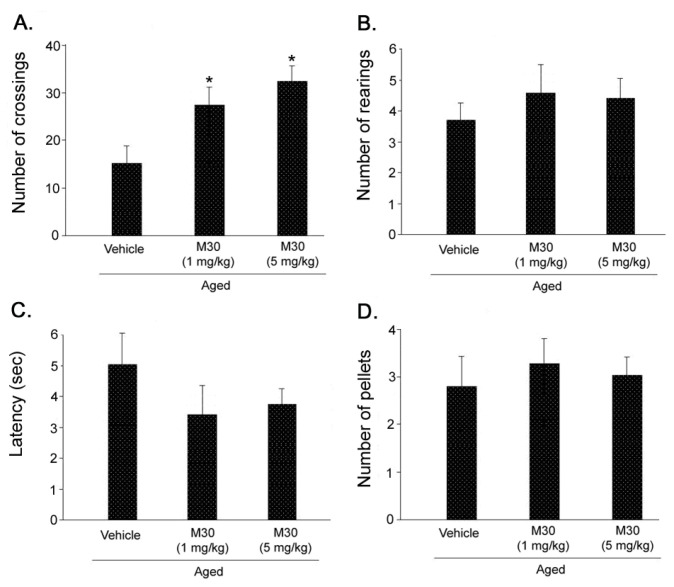
Open field behavior in vehicle- or M30-treated aged mice. Each bar represents the mean ± SEM (n = 7 animals in each group). (**A**) the number of crossings; (**B**) the number of rearings; (**C**) latency to start locomotion; (**D**) the number of fecal pellets. * *p* < 0.05 vs. vehicle-treated aged mice.

**Figure 21 cells-12-00763-f021:**
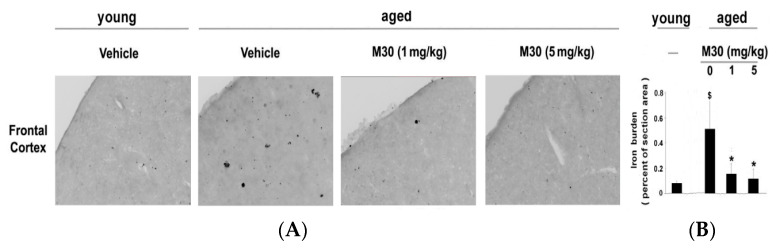
M30 treatment significantly reduced the levels of iron staining in aged mice. (**A**) Iron deposits in cortical and hippocampal regions were detected by enhanced Perl’s iron histochemistry; (**B**) Percentages of Perl’s-positive burdens were calculated by quantitative image analysis. Data are mean ± SEM (n = 3 animals per group; 4 separate fields for each animal). $ *p* < 0.05 vs. vehicle-treated young mice; * *p* < 0.05 vs. vehicle-treated aged mice.

**Figure 22 cells-12-00763-f022:**
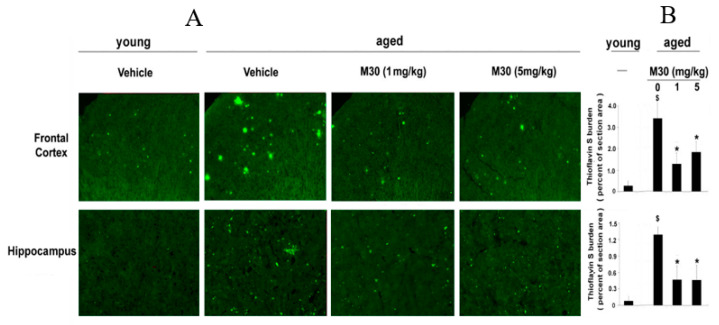
Effect of M30 treatment on fibrillar cerebral amyloidosis in aged mice. (**A**) Immunofluorescence images of mouse brain sections from the indicated regions stained with Thioflavin S; (**B**) Percentages of Thioflavin S-positive burden were calculated by quantitative image analysis. Data are mean ± SEM (n = 3 animals per group; 4 separate fields for each animal). $ *p* < 0.05 vs. vehicle-treated young mice; * *p* < 0.05 vs. vehicle-treated aged mice.

**Figure 23 cells-12-00763-f023:**
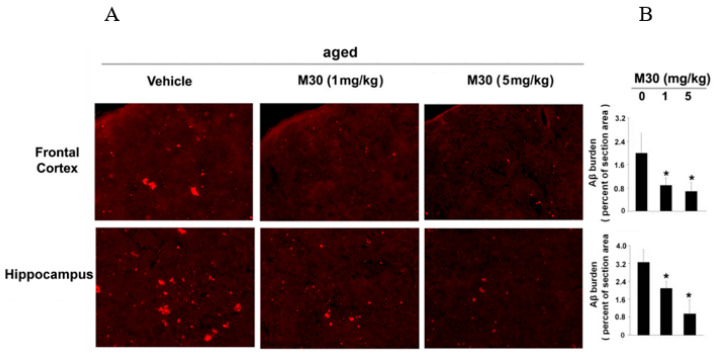
Effect of M30 treatment on cerebral Aβ burden in aged mice. (**A**) Immunohistochemical images showing brain coronal frozen sections from the frontal cortex and hippocampus stained with monoclonal anti-β-Amyloid (4G8) antibody; (**B**) Percentages of Aβ antibody-immunoreactive Aβ plaques were calculated by quantitative image analysis. Data are mean ± SEM (n = 3 animals per group; 4 separate fields for each animal). * *p* < 0.05 vs. vehicle-treated aged mice.

**Figure 24 cells-12-00763-f024:**
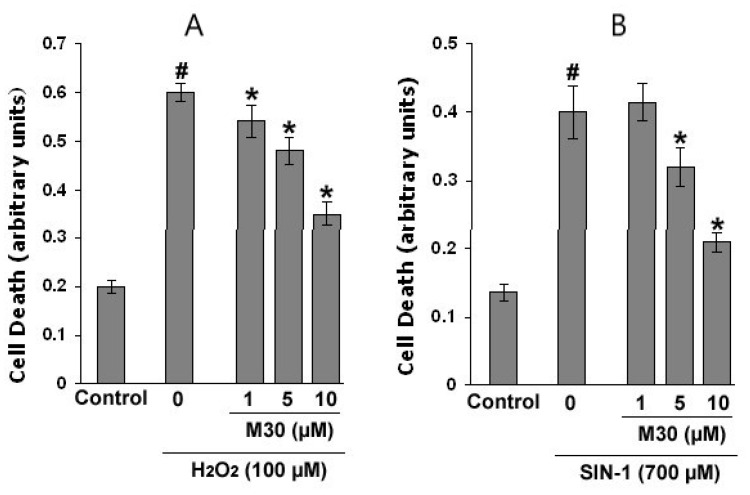
M30 and HLA20 protection of NSC-34 cells against H_2_O_2_- and SIN-1-induced cell death. NSC-34 cells were pretreated without (control) or with M30 or HLA20 (1, 5, and 10 µM) 30 min before exposure to (**A**) H_2_O_2_ (100 µM) and (**B**) SIN-1 (700 µM) for a subsequent 24 h period. Cell death was assayed using apoptotic cell death detection ELISA. Data are expressed as mean ± SEM (n = 6) of a representative experiment that was repeated twice under the same conditions. * *p* < 0.001 vs. NSC-34 cells treated with H_2_O_2_ or SIN-1 only, # *p* < 0.001 vs. control.

**Figure 25 cells-12-00763-f025:**
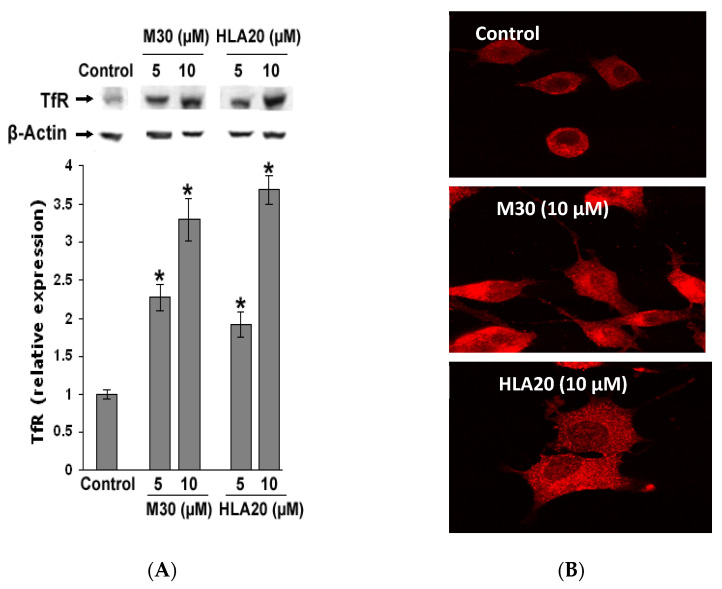
Effect of M30 and HLA20 on transferrin receptor (TfR) protein expression in NSC-34 cell line. (**A**) NSC-34 cells were incubated without (control) or with M30 or HLA20 (5 and 10 µM) for 48 h. TfR expression in cell lysates was examined by immunoblotting analysis. The graphs present densitometric quantifications of the Western blots, expressed as a percentage of control cells, after normalization to β-Actin levels. Results are mean ± SEM values from three independent experiments. * *p* < 0.05, vs. vehicle-treated cells; (**B**) NSC-34 cells were incubated without (control) or with M30 or HLA20 (10 µM) for 48 h. After cell fixation and permeabilization, TfR was detected by confocal microscopy using specific primary antibodies and FITC-conjugated secondary antibodies. The pictures shown are representative of two separate experiments.

**Figure 26 cells-12-00763-f026:**
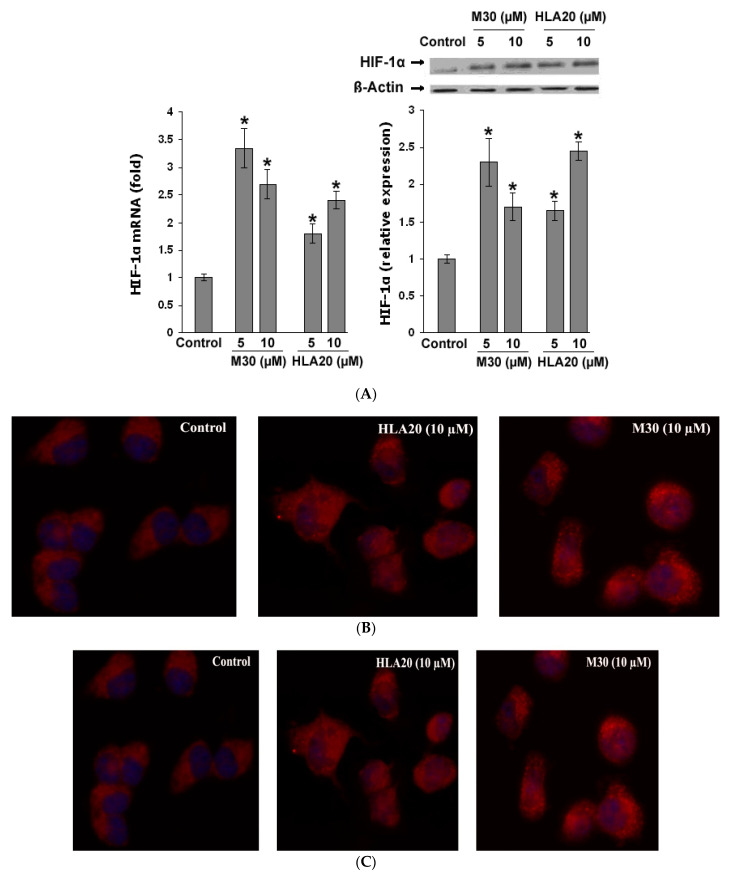
Effect of M30 and HLA20 on hypoxia-inducible factor HIF-1α expression and localization in NSC-34 cells. NSC-34 cells were incubated without (control) or with M30 or HLA20 (5 and 10 µM) for 48 h. (**A**) RNA was extracted and converted to single-stranded for analysis by real-time PCR. The level of HIF-1α relative expression was assessed by normalizing to the housekeeping gene 18S-rRNA. Values are mean ± SEM from two independent experiments conducted in duplicates. * *p* < 0.001 vs. control; (**B**) Western blots showing the expression of (HIF)-1α in NSC-34 cells after incubation with M30 or HLA20 (5 and 10 µM) for 48 h. The graphs represent densitometric quantifications of the Western blots, expressed as a percentage of control cells, after normalization to levels of β-Actin. Results are mean ± SEM values from four independent experiments. * *p* < 0.005 vs. control (vehicle-treated cells); (**C**) After cell fixation and permeabilization, HIF-1α was detected by confocal microscopy using specific primary antibodies and FITC-conjugated secondary antibodies. In control cells, HIF-1α is mostly cytosolic (white arrowhead). In cells treated with M30 and HLA20 (10 µM), it is mainly localized to the cell nucleus (yellow arrowheads). The images are representative fields.

**Figure 27 cells-12-00763-f027:**
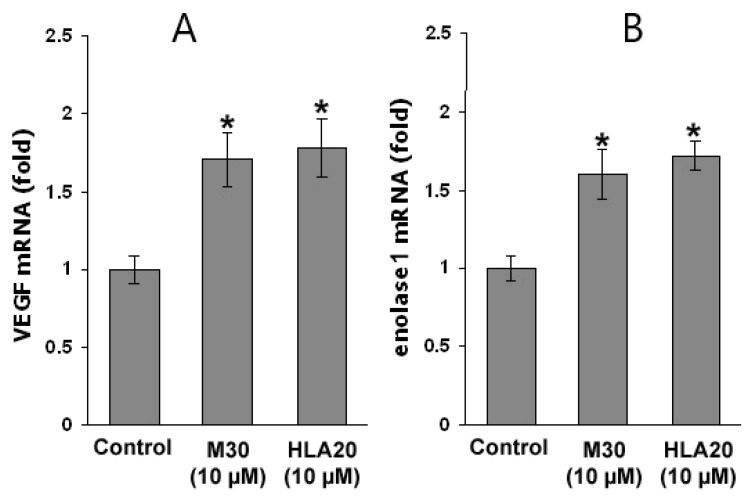
Effect of M30 and HLA20 on gene expression of VEGF and enolase 1 in NSC-34 cells. NSC-34 cells were incubated without (control) or with M30 or HLA20 (10 µM) for 48 h. RNA was extracted and converted to single stranded for analyzing by real-time PCR. Relative expression levels of (**A**) VEGF and (**B**) enolase 1 were assessed by normalizing to the housekeeping gene 18S-rRNA. Values are mean ± SEM from two independent experiments conducted in duplicates. * *p* < 0.05 vs. control.

**Figure 28 cells-12-00763-f028:**
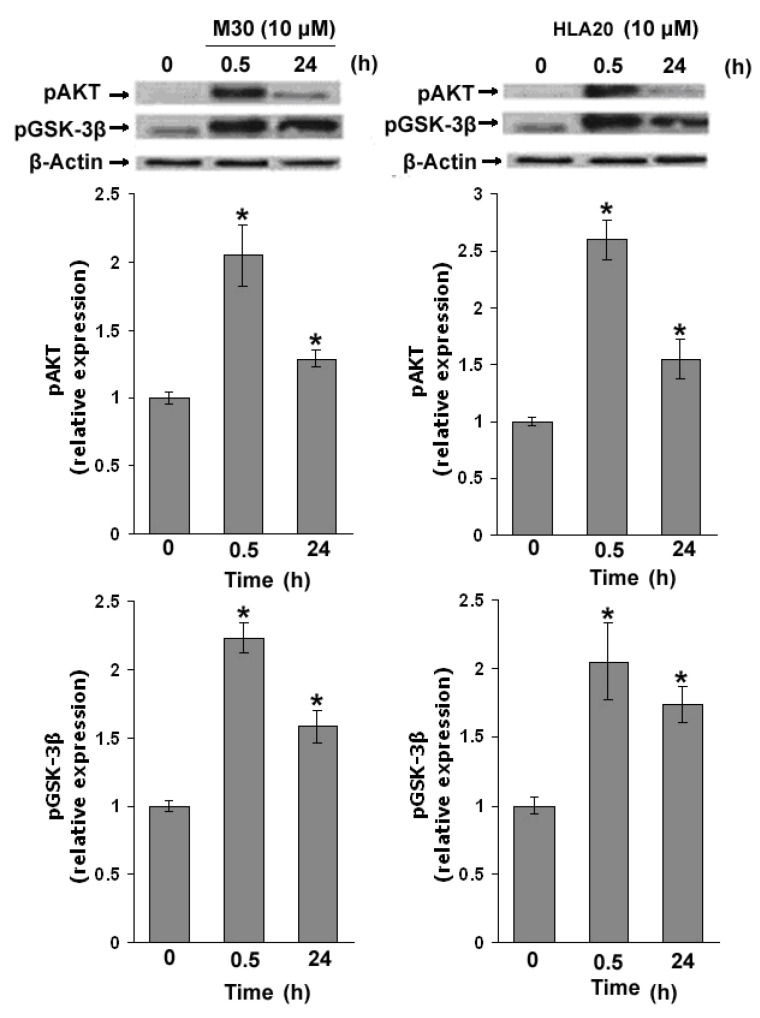
Effect of M30 and HLA20 on phosphorylation of AKT and GSK-3β. NSC-34 cells were incubated without (control) or with M30 (10 µM) and HLA20 (10 µM) for 30 min or 24 h. AKT and GSK-3β activation was evaluated by immunoblotting. The graphs present densitometric quantifications of phosphorylated AKT and GSK-3β blots, normalized for loading using the β-Actin blots. The results are expressed as a percentage of control and are the mean ± SEM (n = 3) of a representative experiment that was repeated twice. * *p* < 0.005 vs. control.

**Figure 29 cells-12-00763-f029:**
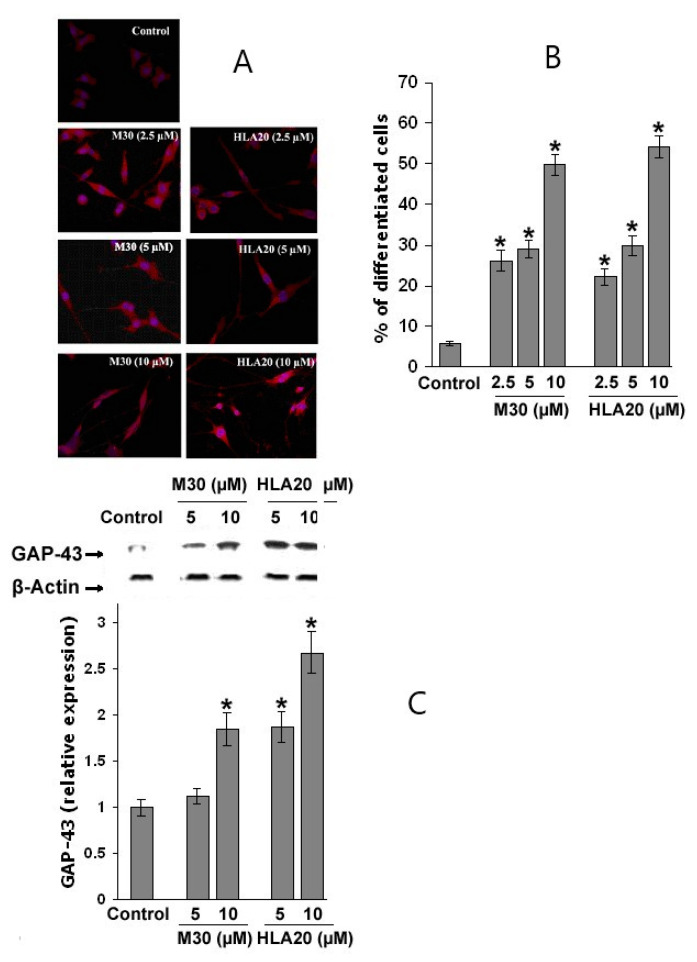
Effect of M30 and HLA20 on NSC-34 neuritogenesis. Cultured NSC-34 cells were deprived of serum for 24 h before exposure to several concentrations of M30 and HLA20 (2.5, 5, and 10 µM). Growth-associated protein (GAP-43) was detected by fluorescence microscopy using a specific primary antibody. The images (**A**) are representative fields from three independent experiments; (**B**) The number of differentiated cells was determined by counting cells with at least one neurite with a length equal to or greater than the diameter of the cell body. A minimum of 250 cells (six to ten separate fields) were examined in a blinded manner, and results are expressed as averages of the differentiated cell percentage (±SEM) detected per sample. * *p* < 0.005 vs. control (vehicle-treated cells); (**C**) Western blots showing the expression of GAP-43 in NSC-34 cells after incubation with several concentrations of M30 and HLA20 (5, 10, and 20 µM) for 48 h. The graphs represent densitometric quantifications of the Western blots, expressed as a percentage of control cells, after normalization to levels of β-Actin.

**Figure 30 cells-12-00763-f030:**
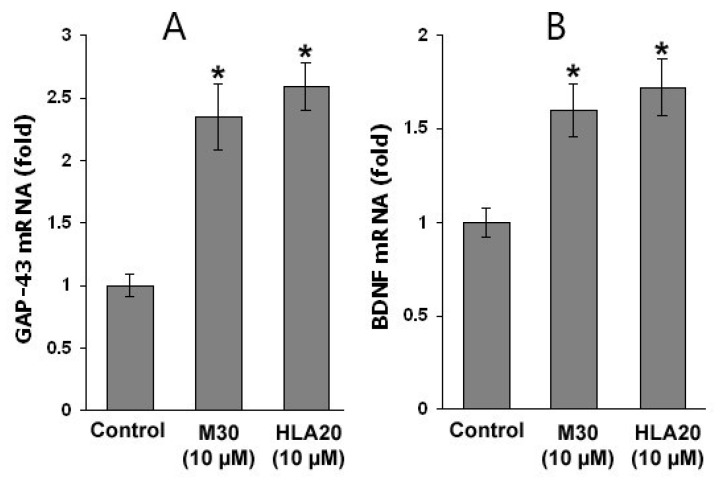
Effect of M30 and HLA20 on gene expression of GAP-43 and BDNF in NSC-34 cells. NSC-34 cells were treated with M30 or HLA20 (10 µM) for 48 h. RNA was extracted and converted to single-stranded for analysis by real-time PCR. Relative expression levels of (**A**) GAP-43 and (**B**) BDNF were assessed by normalizing to the housekeeping gene 18S-rRNA. Values are mean ± SEM from two independent experiments conducted in duplicates. Results are mean ± SEM values from four independent experiments. * *p* < 0.05 vs. control (vehicle-treated cells).

**Figure 31 cells-12-00763-f031:**
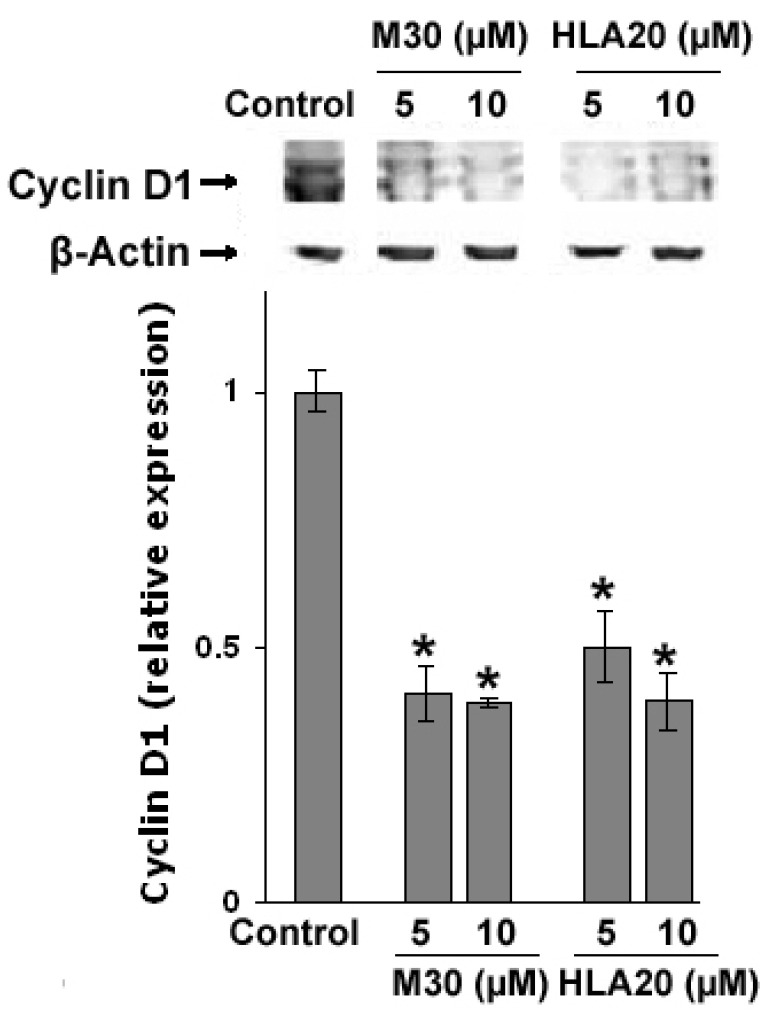
Effect of M30 and HLA20 on Cyclin D1 expression in NSC-34 cells. NSC-34 cells were grown in the presence of M30 or HLA20 (5 and 10 µM) in a serum-free medium for 48 h. Whole-cell lysates were separated by SDS-PAGE and immunoblotted with Cyclin D1 antibody. The graphs represent densitometric quantifications of the Western blots, expressed as a percentage of control cells (vehicle-treated cells), after normalization to levels of β-Actin. Results are mean ± SEM values from two independent experiments. * *p* < 0.001 vs. control.

**Figure 32 cells-12-00763-f032:**
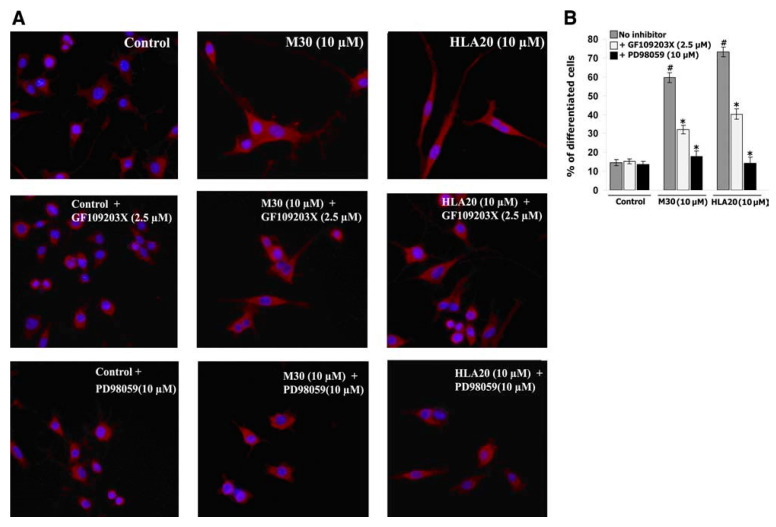
Attenuation of the neuritogenic effect of M30 and HLA20 by specific inhibitors of PKC and MAPK/ERK kinase (MEK). NSC-34 cells were incubated with PKC inhibitor GF109203X (2.5 µM) or MEK inhibitor PD98059 (10 µM) for 1 h before administration of M30 (10 µM) or HLA20 (10 µM) for a further 24 h. The cells were fixed and permeabilized for GAP-43 detection. (**A**) The images are representative fields from three independent experiments; (**B**) the histogram represents averages of the differentiated cell percentages (±SEM). One-way ANOVA followed by Student’s *t*-test was used for statistical analysis. * *p* < 0.001 vs. respective controls; # *p* < 0.001 vs. M30 and HLA20 only (without inhibitors) treated cells.

**Figure 33 cells-12-00763-f033:**
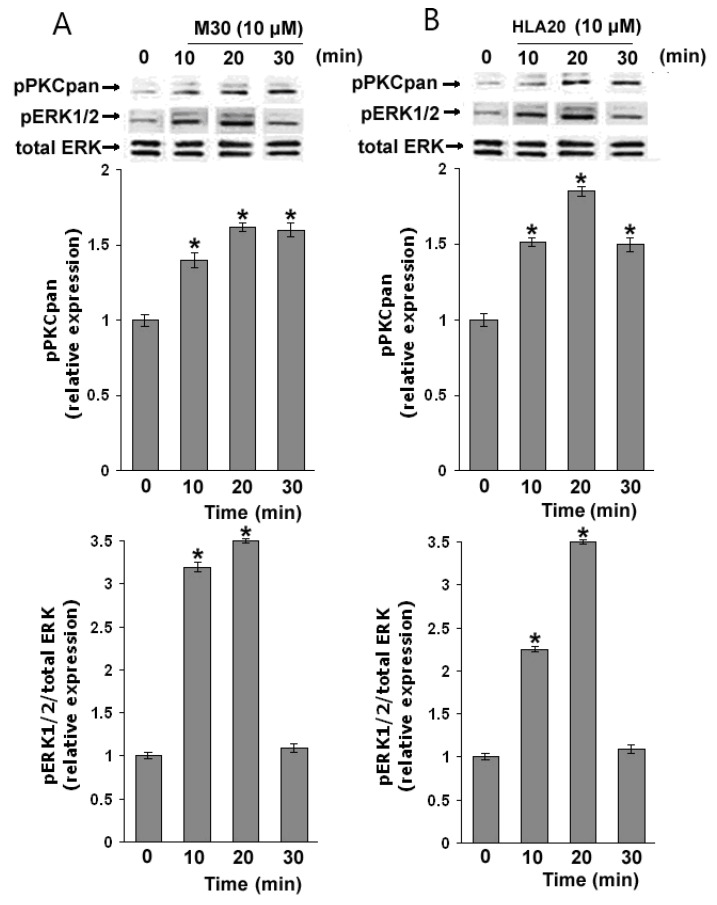
Effect of M30 and HLA20 on phosphorylation of PKCpan and ERK1/2. NSC-34 cells were incubated without (control) or with (**A**) M30 (10 µM) or (**B**) HLA20 (10 µM) for increasing intervals (10, 20, and 30 min). pPKCpan and ERK1/2 activation were evaluated by immunoblotting. The graphs present densitometric quantifications of phosphorylated PKCpan and ERK1/2 blots normalized to total ERK protein blots. The results are expressed as a percentage of control and are the mean ± SEM (n = 3) of the representative experiment that was repeated twice. * *p* < 0.005 vs. control.

**Figure 34 cells-12-00763-f034:**
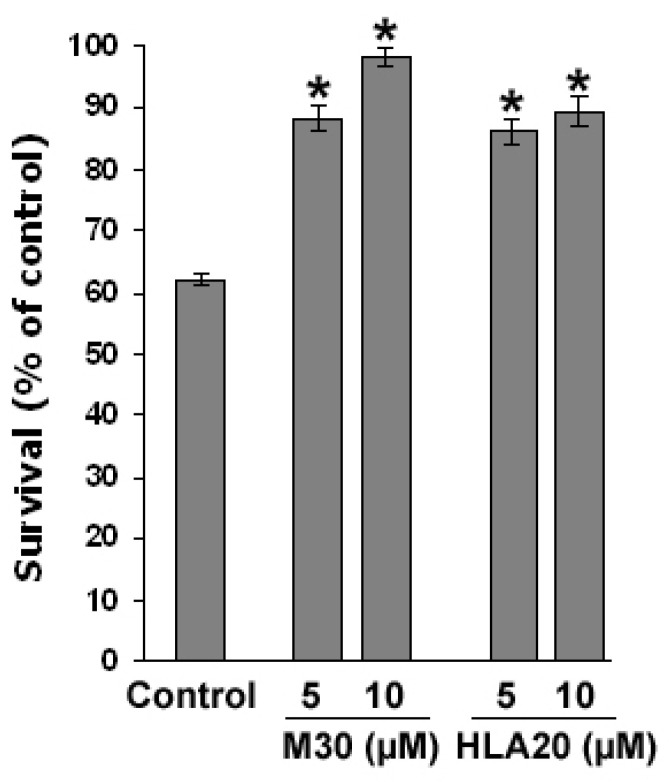
Effect of M30 and HLA20 on viability of G93ASOD1 NSC-34. NCS-34 cells expressing G93A-SOD1 mutation were treated with M30 and HLA20 (5 and 10 M) for 48 h.Empty vector-transfected vehicle-treated cells served as control. Cell viability was evaluated by MTT assay. * *p* < 0.05.

**Figure 35 cells-12-00763-f035:**
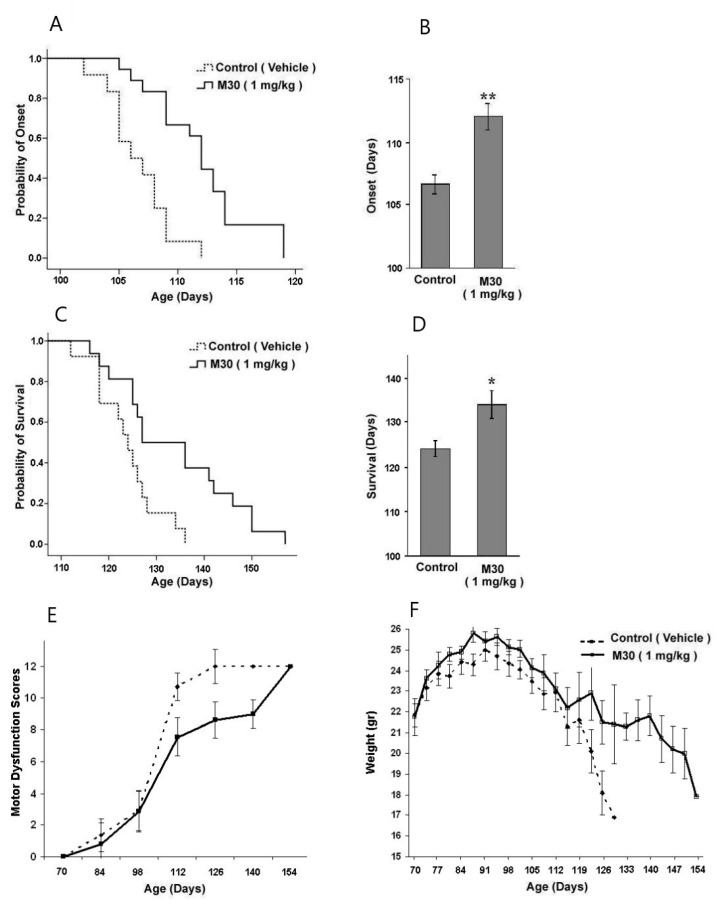
Effect of M30 treatment on motor dysfunction onset, survival time, motor deficits and weight in G93A-SOD1 mutant ALS transgenic mice. Mutant G93A-SOD1 mice were treated by the oral gavage method with vehicle (control) or M30 (1 mg/kg) four times a week starting from the 70th day after birth and continuing until death. Plots present cumulative probability of (**A**) the onset of the symptoms (n = 13–16 per group; *p* < 0.001; log-rank Mantel–Cox test) and (**C**) overall survival (n = 13–16 per group; *p* < 0.025; log-rank Mantel–Cox test) against the age of the mutant mice. Histograms present (**B**) mean onset (days) and (**D**) mean survival (days) of vehicle- or M30-treated G93A-SOD1 mice. Values are means ± SEM (n = 13–16 per group; * *p* < 0.05, ** *p* < 0.001 vs. control group; one-way ANOVA). (**E**) Overall neurological deficit scores vs. the age of animals. The total neurological deficits were determined from four independent tests (rotarod performance, postural reflex, screen grasping, and tail suspension behavior), as described in Materials and Methods. The total score of 12 represents a complete loss of motor function. Values are means ± SEM (n = 13–16 per group). (**F**) Weight vs. the age of animals. Values are means ± SEM.

**Figure 36 cells-12-00763-f036:**
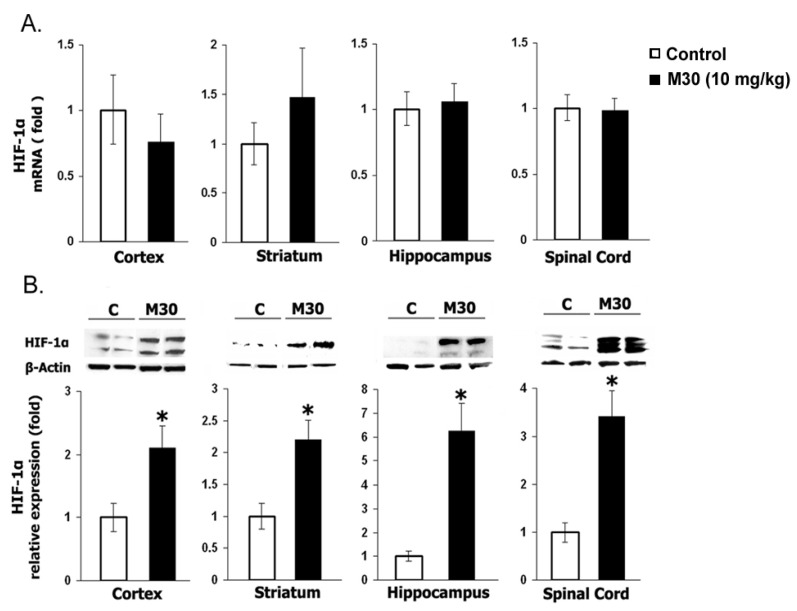
Effect of M30 on HIF-1α expression levels in mouse cortex, hippocampus, and spinal cord following M30 administration. Adult C57BL/6 mice were randomly assigned to vehicle (n = 11) or M30 (n = 12) (10 mg/kg/day for 30 days). Treatment was initiated at the age of 28 days old by the oral gavage method. (**A**) HIF-1α gene expression was measured by quantitative real-time PCR. The amount of each product was normalized to the housekeeping gene 18S-rRNA. Data are expressed as relative gene expression vs. control (fold) and represent means ± SEM; (**B**) HIF-1α protein level was examined by immunoblotting analysis in cell lysates. The loading of the lanes was normalized to levels of β-Actin. Western blotting is representative, and the densitometric analysis is defined as relative protein levels (fold vs. control). * *p* < 0.05 vs. control (vehicle-treated mice).

**Figure 37 cells-12-00763-f037:**
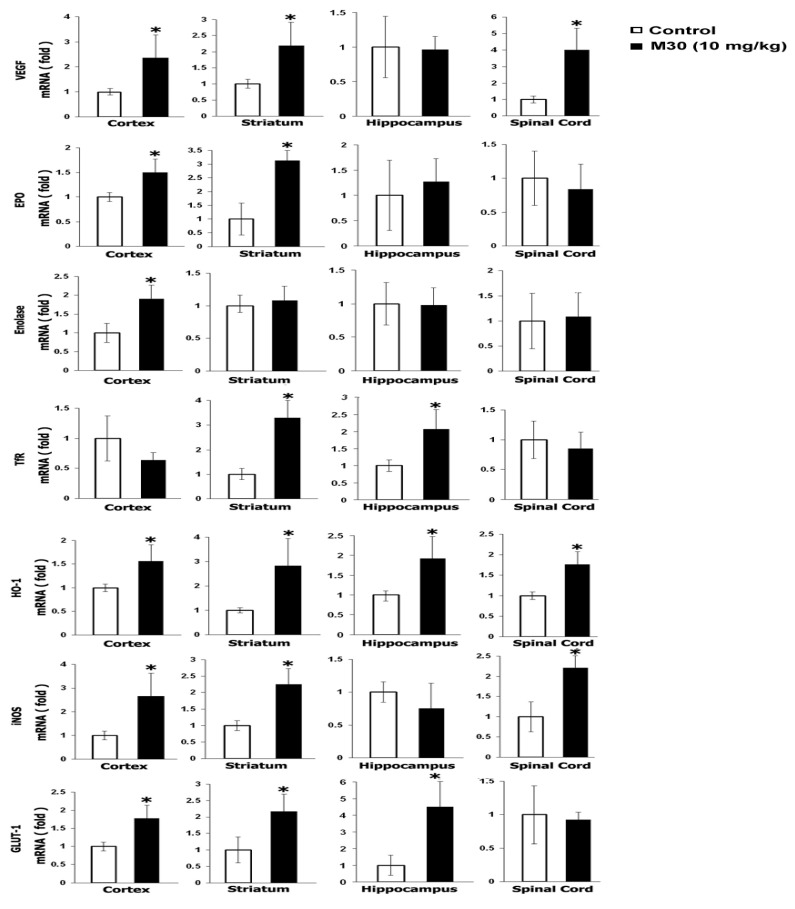
M30 regulates the expression of HIF-1-related genes in the mouse cortex, hippocampus, striatum, and spinal cord. Adult mice were treated as described in Figure 36. The expression levels of VEGF, EPO, enolase-1, TfR, HO-1, iNOS, and GLUT-1 were measured by quantitative real-time RT-PCR. The amount of each product was normalized to the housekeeping gene 18S-rRNA. Data are expressed as relative gene expression vs. the respective control (fold) and represent means ± SEM. * *p* < 0.05 vs. control (vehicle-treated mice).

**Figure 38 cells-12-00763-f038:**
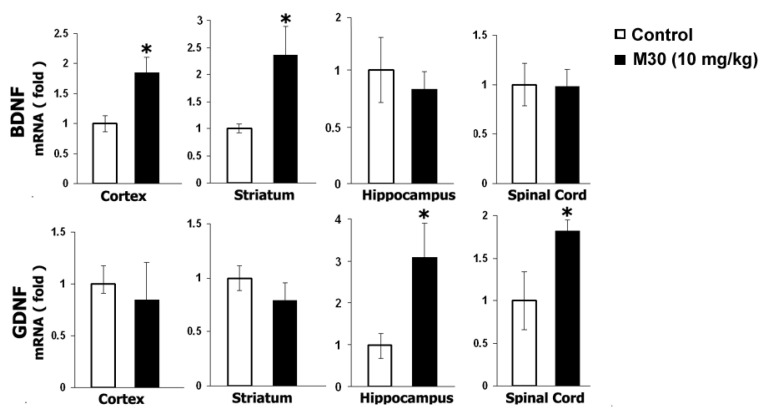
BDNF and GDNF mRNA levels normalized to the housekeeping gene 18S-rRNA in mouse CNS in response to M30 compared to controls. Adult mice were treated as described in Figure 36. The amount of each product was normalized to the housekeeping gene 18S-rRNA. Data are expressed as relative gene expression vs. control (fold) and represent means ± SEM. * *p* < 0.05 vs. control (vehicle-treated mice).

**Figure 39 cells-12-00763-f039:**
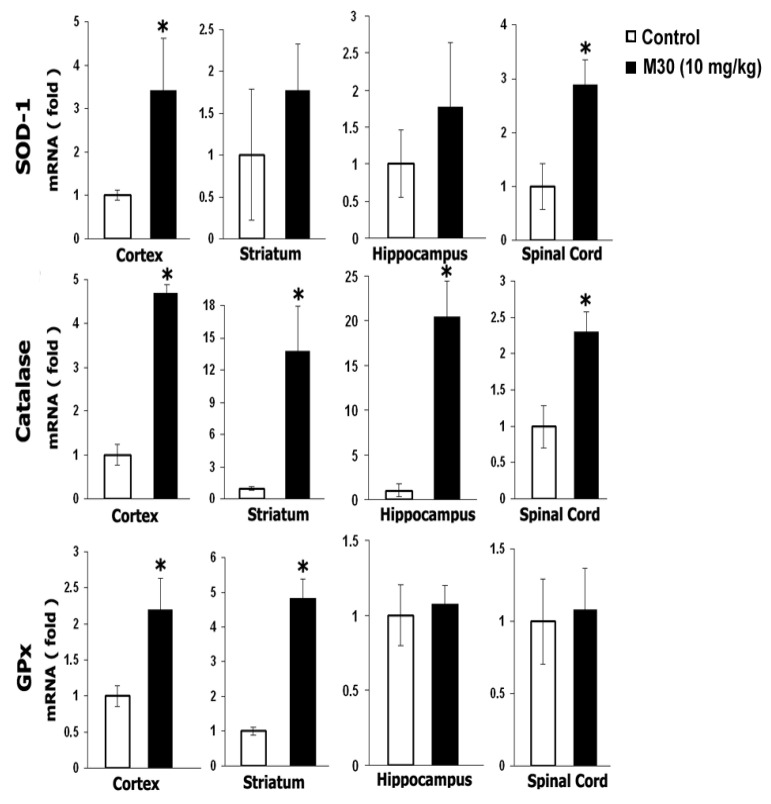
SOD-1, catalase, and GPx gene expression in mouse CNS in response to M30. Adult mice were treated as described in Figure 36. Data are expressed as relative gene expression vs. control (fold) and represent means ± SEM (n = 11–12 mice for each group). * *p* < 0.05 vs. control (vehicle-treated mice).

**Figure 40 cells-12-00763-f040:**
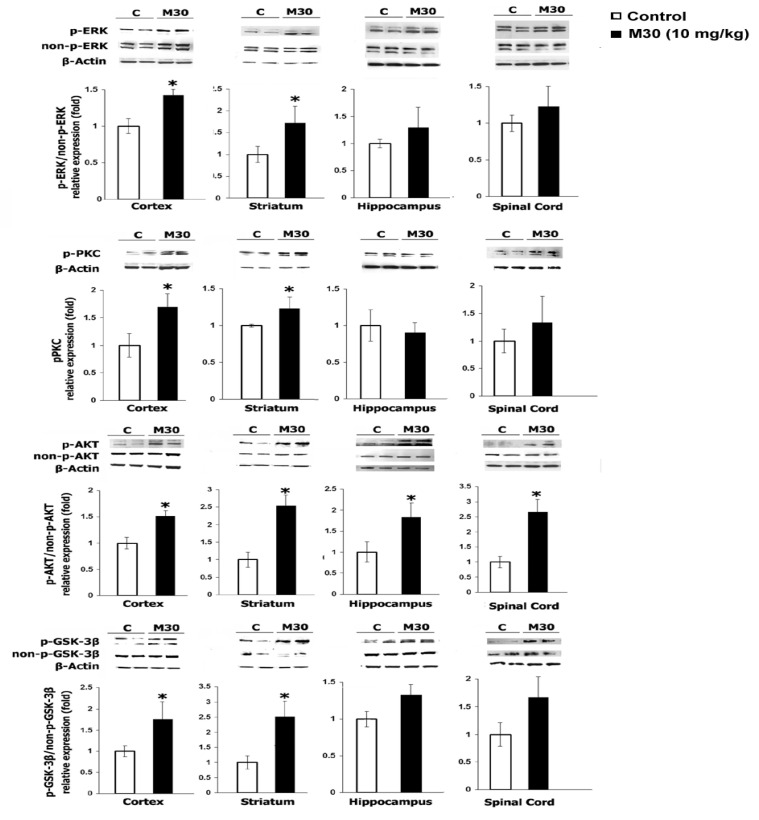
Effect of M30 treatment on the phosphorylation of PKC, ERK1/2, AKT, and GSK-3β in cortex, hippocampus, striatum, and spinal cord. Adult mice were treated as described in Figure 36. Phosphorylation of PKC, ERK1/2, AKT (Ser-473), and GSK-3β (Ser-9) was evaluated by immunoblotting. The loading of the lanes was normalized to the levels of β-Actin. Results are means ± SEM. * *p* < 0.05 vs. control (vehicle-treated mice).

**Figure 41 cells-12-00763-f041:**
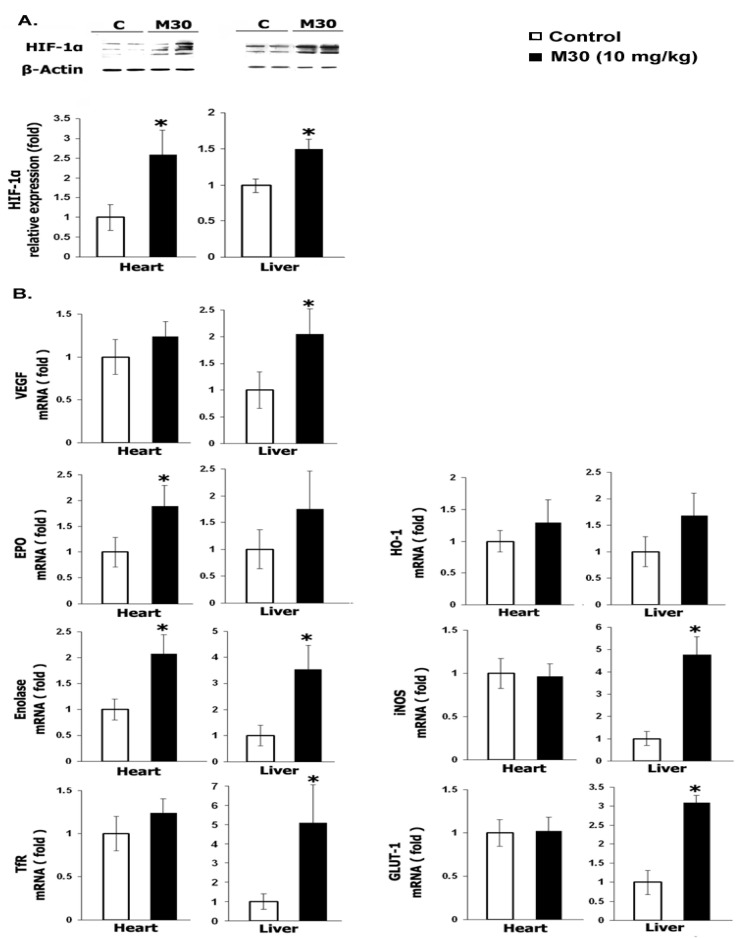
Effects of M30 on (**A**) HIF-1α expression; (**B**) HIF-1-related genes; (**C**) antioxidant enzymes gene expression; and (**D**) phosphorylation of PKC, ERK1/2, AKT, and GSK-3β in heart and liver. Adult mice were treated as described in Figure 36. Data are means ± SEM (n = 11–12 mice for each group). * *p* < 0.05 vs. control (vehicle-treated mice).

**Figure 42 cells-12-00763-f042:**
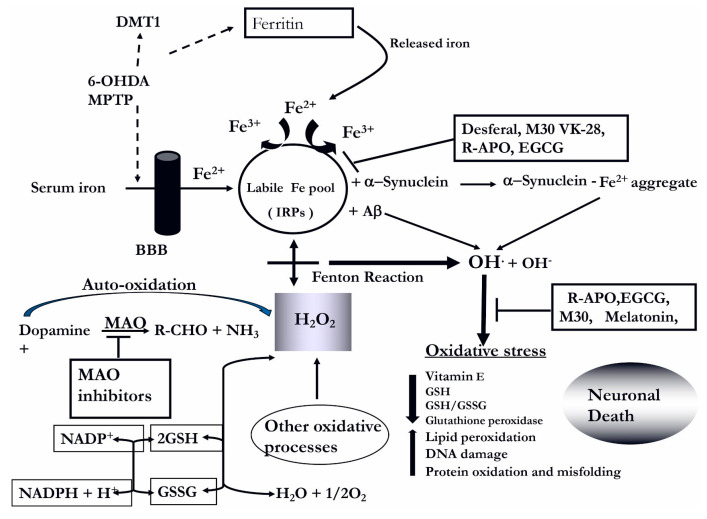
Schematic diagram illustrating the potential neuroprotective effects of iron chelators and radical scavengers on attenuation of oxidative stress. Dopamine oxidation, through MAO-B, leads to elevated levels of H_2_O_2_, which in turn, participates in Fenton chemistry, thus creating a vicious circle of oxidative damage. MAO-B inhibitors can inhibit the generation of H_2_O_2_ and subsequently prevent neuronal damage. An additional contribution to oxidative stress stems from excessive iron accumulation, which can cause damage through the release of reactive oxygen species via the Fenton reaction. Ferrous iron (Fe^2^+) is oxidized to ferric iron (Fe^3^+) through a reaction with hydrogen peroxide (H_2_O_2_). The by-products of this reaction are the hydroxyl anion (OH^−^) and the hydroxyl radical (OH). Consequently, the antioxidant GSH is significantly reduced, resulting in cell death. Iron chelators and radical scavengers can inhibit the Fenton reaction and enhance antioxidant activity.

**Table 1 cells-12-00763-t001:** Parameters scored during the modified SHIRPA Screen test in vehicle- or M30-treated aged mice.

Task Name	Vehicle	M30 (1 mg/kg)	M30 (5 mg/kg)
Transfer activity	1 (0–2)	0 (0–0)	0 (0–0)
Locomotor activity	1 (0–2)	1 (0–1)	0 (0–0)
Positional passivity	1 (0–2)	0 (0–1)	0 (0–0)
Touch escape	1 (0–2)	0 (0–0)	0 (0–0)
Vocalization	1 (0–2)	0 (0–1)	0 (0–0)

**Table 2 cells-12-00763-t002:** Effect of M30 on MAO-A and -B inhibition in the cerebellum of aged mice.

Enzyme Inhibition	M30 (1 mg/kg)	M30 (5 mg/kg)
MAO-A (% of control)	7.5 ± 2.9	35.2 ± 1.5 *
MAO-B (% of control)	7.4 ± 11.8	37.1 ± 7.3 *

MAO-A and -B activities were determined, as described in Materials and Methods. Results represent the percentage of inhibition of control activities, mean ± SEM (n = 6–8 animals per group). Vehicle-treated aged mice served as the control group. * *p* < 0.05 vs. vehicle-treated aged mice.

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
