# Peer review of "The Neuroprotective Activities of the Novel Multi-Target Iron-Chelators in Models of Alzheimer’s Disease, Amyotrophic Lateral Sclerosis and Aging"

_cells, 2023, doi:10.3390/cells12050763_

Round 1

Reviewer 1 Report

In this work, the authors review the activity of two of their iron chelators in multiple models of neurodegenerative disorders. The review is well-organized and easy to follow. It addresses an important aspect of iron’s role in neurodegenerative disorders.

I find 250 citations a bit excessive, and I urge the authors to reduce that number. More importantly, the authors cited 32 of Dr. Youdim’s previous articles. From this point of view, the review reads more like an advertisement than a critical overview of the literature. Along these lines, it would be appropriate to also describe the effects of other iron chelators to make this a more comprehensive review. 

Author Response

Reviewer 1’s comments and authors’ responses

“In this work, the authors review the activity of two of their iron chelators in multiple models of neurodegenerative disorders. The review is well-organized and easy to follow. It addresses an important aspect of iron’s role in neurodegenerative disorders.”

Response: Many thanks for Reviewer 1’s encouraging comments.

“I find 250 citations a bit excessive, and I urge the authors to reduce that number. More importantly, the authors cited 32 of Dr. Youdim’s previous articles. From this point of view, the review reads more like an advertisement than a critical overview of the literature. Along these lines, it would be appropriate to also describe the effects of other iron chelators to make this a more comprehensive review.” 

Response: I started the work on brain iron metabolism in 1974 when no one paid attention to it. Our lab was also the first to use iron chelators as neuroprotective drugs in models of Parkinson's disease (PD) and developed novel multi-target brain permeable iron chelators for PD and AD. This review manuscript was an invitation from the guest editor of this special issue- Dr. Xudong Huang. Dr. Huang has specifically asked me to focus on the significant original works we did in developing multi-target iron chelators for neurodegenerative diseases. We thus limit our review scope to our research work per Dr. Huang's request. If he had asked me to review the field of iron and AD, I would have done so.

Reviewer 2 Report

Dear Authors:

The manuscript "The Neuroprotective Activities of the Novel Multi-Target Iron-Chelators in Models of Alzheimer’s Disease, Amyotrophic Lateral Sclerosis, and Aging " by Kupershmidt et al has demonstrated multifunctional iron-chelating compounds can upregulate several neuroprotective-adaptive mechanisms and pro-survival signaling pathways in the brain and might function as ideal drugs for neurodegenerative disorders, such as PD, AD, ALS, and aging-related cognitive decline, in which oxidative stress and iron-mediated toxicity and dysregulation of iron homeostasis have been implicated. I have just a few suggestions. 

Some background information or references are missing. In introduction, about the pathogenic factors in neurodegenerative diseases, vascular dysfunction is also a great potential factor, which leads brain ischemia. Many articles and reviews demonstrated it. (Please cite: From 1901 to 2022, how far are we from truly understanding the pathogenesis of age-related dementia? Geroscience. 2022 Jun;44(3):1879-1883. doi: 10.1007/s11357-022-00591-7.)

Best,

Author Response

Reviewer 2’s comments and authors’ responses

“The manuscript "The Neuroprotective Activities of the Novel Multi-Target Iron-Chelators in Models of Alzheimer’s Disease, Amyotrophic Lateral Sclerosis, and Aging " by Kupershmidt et al has demonstrated multifunctional iron-chelating compounds can upregulate several neuroprotective-adaptive mechanisms and pro-survival signaling pathways in the brain and might function as ideal drugs for neurodegenerative disorders, such as PD, AD, ALS, and aging-related cognitive decline, in which oxidative stress and iron-mediated toxicity and dysregulation of iron homeostasis have been implicated. I have just a few suggestions. 

Some background information or references are missing. In introduction, about the pathogenic factors in neurodegenerative diseases, vascular dysfunction is also a great potential factor, which leads brain ischemia. Many articles and reviews demonstrated it. (Please cite: From 1901 to 2022, how far are we from truly understanding the pathogenesis of age-related dementia? Geroscience. 2022 Jun;44(3):1879-1883. doi: 10.1007/s11357-022-00591-7.)

 Best,”

Response: I want to thank this reviewer for reviewing our review manuscript. I was invited by the guest editor- Dr. Xudong Huang, to contribute a review of the significant original works we did in developing multi-target iron chelators for neurodegenerative diseases. Dr. Huang has specifically asked me to focus on iron chelators and neurodegenerative diseases from my lab. Therefore, I limit my review scope to my own research work. We believe that brain ischemia is beyond the scope of this review. However, we’ve mentioned it in the Introduction section and cited the reference (see lines 31-33) as the reviewer has suggested.

Round 2

Reviewer 1 Report

I am satisfied by their response to original comment